# Direct Preference Optimization:
# Your Language Model is Secretly a Reward Model

**Rafael Rafailov**[*†]    **Archit Sharma**[*†]    **Eric Mitchell**[*†]

**Stefano Ermon**[†‡]    **Christopher D. Manning**[†]    **Chelsea Finn**[†]

[†]Stanford University  [‡]CZ Biohub
{rafailov,architsh,eric.mitchell}@cs.stanford.edu

## Abstract

While large-scale unsupervised language models (LMs) learn broad world knowledge and some reasoning skills, achieving precise control of their behavior is difficult due to the completely unsupervised nature of their training. Existing methods for gaining such steerability collect human labels of the relative quality of model generations and fine-tune the unsupervised LM to align with these preferences, often with reinforcement learning from human feedback (RLHF). However, RLHF is a complex and often unstable procedure, first fitting a reward model that reflects the human preferences, and then fine-tuning the large unsupervised LM using reinforcement learning to maximize this estimated reward without drifting too far from the original model. In this paper, we leverage a mapping between reward functions and optimal policies to show that this constrained reward maximization problem can be *optimized exactly* with a single stage of policy training, essentially solving a classification problem on the human preference data. The resulting algorithm, which we call *Direct Preference Optimization* (DPO), is stable, performant, and computationally lightweight, eliminating the need for fitting a reward model, sampling from the LM during fine-tuning, or performing significant hyperparameter tuning. Our experiments show that DPO can fine-tune LMs to align with human preferences as well as or better than existing methods. Notably, fine-tuning with DPO exceeds RLHF's ability to control sentiment of generations and improves response quality in summarization and single-turn dialogue while being substantially simpler to implement and train.

## 1 Introduction

Large unsupervised language models (LMs) trained on very large datasets acquire surprising capabilities [11, 7, 40, 8]. However, these models are trained on data generated by humans with a wide variety of goals, priorities, and skillsets. Some of these goals and skillsets may not be desirable to imitate; for example, while we may want our AI coding assistant to *understand* common programming mistakes in order to correct them, nevertheless, when generating code, we would like to bias our model toward the (potentially rare) high-quality coding ability present in its training data. Similarly, we might want our language model to be *aware* of a common misconception believed by 50% of people, but we certainly do not want the model to claim this misconception to be true in 50% of queries about it! In other words, selecting the model's *desired responses and behavior* from its very wide *knowledge and abilities* is crucial to building AI systems that are safe, performant, and controllable [26]. While existing methods typically steer LMs to match human preferences using reinforcement learning (RL),

---

[*]Equal contribution; more junior authors listed earlier.

37th Conference on Neural Information Processing Systems (NeurIPS 2023).

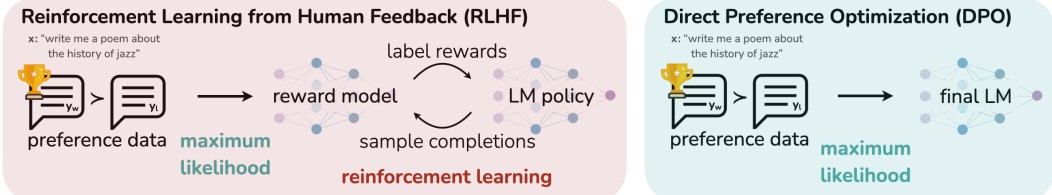

Figure 1: **DPO optimizes for human preferences while avoiding reinforcement learning.** Existing methods for fine-tuning language models with human feedback first fit a reward model to a dataset of prompts and human preferences over pairs of responses, and then use RL to find a policy that maximizes the learned reward. In contrast, DPO directly optimizes for the policy best satisfying the preferences with a simple classification objective, without an explicit reward function or RL.

we will show that the RL-based objective used by existing methods can be optimized exactly with a simple binary cross-entropy objective, greatly simplifying the preference learning pipeline.

At a high level, existing methods instill the desired behaviors into a language model using curated sets of human preferences representing the types of behaviors that humans find safe and helpful. This preference learning stage occurs after an initial stage of large-scale unsupervised pre-training on a large text dataset. While the most straightforward approach to preference learning is supervised fine-tuning on human demonstrations of high quality responses, the most successful class of methods is reinforcement learning from human (or AI) feedback (RLHF/RLAIF; [12, 2]). RLHF methods fit a reward model to a dataset of human preferences and then use RL to optimize a language model policy to produce responses assigned high reward without drifting excessively far from the original model. While RLHF produces models with impressive conversational and coding abilities, the RLHF pipeline is considerably more complex than supervised learning, involving training multiple LMs and sampling from the LM policy in the loop of training, incurring significant computational costs.

In this paper, we show how to directly optimize a language model to adhere to human preferences, without explicit reward modeling or reinforcement learning. We propose *Direct Preference Optimization (DPO)*, an algorithm that implicitly optimizes the same objective as existing RLHF algorithms (reward maximization with a KL-divergence constraint) but is simple to implement and straightforward to train. Intuitively, the DPO update increases the relative log probability of preferred to dispreferred responses, but it incorporates a dynamic, per-example importance weight that prevents the model degeneration that we find occurs with a naive probability ratio objective. Like existing algorithms, DPO relies on a theoretical preference model (such as the Bradley-Terry model; [5]) that measures how well a given reward function aligns with empirical preference data. However, while existing methods use the preference model to define a preference loss to train a reward model and then train a policy that optimizes the learned reward model, DPO uses a change of variables to define the preference loss as a function of the policy directly. Given a dataset of human preferences over model responses, DPO can therefore optimize a policy using a simple binary cross entropy objective, without explicitly learning a reward function or sampling from the policy during training.

Our main contribution is Direct Preference Optimization (DPO), a simple RL-free algorithm for training language models from preferences. Our experiments show that DPO is at least as effective as existing methods, including PPO-based RLHF, for learning from preferences in tasks such as sentiment modulation, summarization, and dialogue, using language models with up to 6B parameters.

## 2   Related Work

Self-supervised language models of increasing scale learn to complete some tasks zero-shot [31] or with few-shot prompts [6, 25, 11]. However, their performance on downstream tasks and alignment with user intent can be significantly improved by fine-tuning on datasets of instructions and human-written completions [23, 36, 13, 39]. This 'instruction-tuning' procedure enables LLMs to generalize to instructions outside of the instruction-tuning set and generally increase their usability [13]. Despite the success of instruction tuning, *relative* human judgments of response quality are often easier to collect than expert demonstrations, and thus subsequent works have fine-tuned LLMs with datasets of human preferences, improving proficiency in translation [18], summarization [38, 48], story-telling [48], and instruction-following [26, 32]. These methods first optimize a neural network reward function for compatibility with the dataset of preferences under a preference model such as the

Bradley-Terry model [5], then fine-tune a language model to maximize the given reward using reinforcement learning algorithms, commonly REINFORCE [44], proximal policy optimization (PPO; [37]), or variants [32]. A closely-related line of work leverages LLMs fine-tuned for instruction following with human feedback to generate additional synthetic preference data for targeted attributes such as safety or harmlessness [2], using only weak supervision from humans in the form of a text rubric for the LLM's annotations. These methods represent a convergence of two bodies of work: one body of work on training language models with reinforcement learning for a variety of objectives [33, 27, 45] and another body of work on general methods for learning from human preferences [12, 19]. Despite the appeal of using relative human preferences, fine-tuning large language models with reinforcement learning remains a major practical challenge; this work provides a theoretically-justified approach to optimizing relative preferences without RL.

Outside of the context of language, learning policies from preferences has been studied in both bandit and reinforcement learning settings, and several approaches have been proposed. Contextual bandit learning using preferences or rankings of actions, rather than rewards, is known as a contextual dueling bandit (CDB; [47, 14]). In the absence of absolute rewards, theoretical analysis of CDBs substitutes the notion of an optimal policy with a *von Neumann winner*, a policy whose expected win rate against *any* other policy is at least 50% [14]. However, in the CDB setting, preference labels are given online, while in learning from human preferences, we typically learn from a fixed batch of offline preference-annotated action pairs [46]. Similarly, *preference-based RL* (PbRL) learns from binary preferences generated by an *unknown* 'scoring' function rather than rewards [9, 35]. Various algorithms for PbRL exist, including methods that can reuse off-policy preference data, but generally involve first explicitly estimating the latent scoring function (i.e. the reward model) and subsequently optimizing it [16, 9, 12, 34, 19]. We instead present a single stage policy learning approach that directly optimizes a policy to satisfy preferences.

## 3 Preliminaries

We review the RLHF pipeline in Ziegler et al. (and later [38, 1, 26]). It usually includes three phases: 1) supervised fine-tuning (SFT); 2) preference sampling and reward learning and 3) RL optimization.

**SFT**: RLHF typically begins by fine-tuning a pre-trained LM with supervised learning on high-quality data for the downstream task(s) of interest (dialogue, summarization, etc.), to obtain a model $\pi^{\text{SFT}}$.

**Reward Modelling Phase**: In the second phase the SFT model is prompted with prompts $x$ to produce pairs of answers $(y_1, y_2) \sim \pi^{\text{SFT}}(y \mid x)$. These are then presented to human labelers who express preferences for one answer, denoted as $y_w \succ y_l \mid x$ where $y_w$ and $y_l$ denotes the preferred and dispreferred completion amongst $(y_1, y_2)$ respectively. The preferences are assumed to be generated by some latent reward model $r^*(y, x)$, which we do not have access to. There are a number of approaches used to model preferences, the Bradley-Terry (BT) [5] model being a popular choice (although more general Plackett-Luce ranking models [30, 21] are also compatible with the framework if we have access to several ranked answers). The BT model stipulates that the human preference distribution $p^*$ can be written as:

$$p^*(y_1 \succ y_2 \mid x) = \frac{\exp\left(r^*(x, y_1)\right)}{\exp\left(r^*(x, y_1)\right) + \exp\left(r^*(x, y_2)\right)}. \tag{1}$$

Assuming access to a static dataset of comparisons $\mathcal{D} = \left\{x^{(i)}, y_w^{(i)}, y_l^{(i)}\right\}_{i=1}^N$ sampled from $p^*$, we can parametrize a reward model $r_\phi(x, y)$ and estimate the parameters via maximum likelihood. Framing the problem as a binary classification we have the negative log-likelihood loss:

$$\mathcal{L}_R(r_\phi, \mathcal{D}) = -\mathbb{E}_{(x, y_w, y_l) \sim \mathcal{D}}\left[\log \sigma(r_\phi(x, y_w) - r_\phi(x, y_l))\right] \tag{2}$$

where $\sigma$ is the logistic function. In the context of LMs, the network $r_\phi(x, y)$ is often initialized from the SFT model $\pi^{\text{SFT}}(y \mid x)$ with the addition of a linear layer on top of the final transformer layer that produces a single scalar prediction for the reward value [48]. To ensure a reward function with lower variance, prior works normalize the rewards, such that $\mathbb{E}_{x, y \sim \mathcal{D}}\left[r_\phi(x, y)\right] = 0$ for all $x$.

**RL Fine-Tuning Phase**: During the RL phase, we use the learned reward function to provide feedback to the language model. In particular, we formulate the following optimization problem

$$\max_{\pi_\theta} \mathbb{E}_{x \sim \mathcal{D}, y \sim \pi_\theta(y|x)}\left[r_\phi(x, y)\right] - \beta \mathbb{D}_{\text{KL}}\left[\pi_\theta(y \mid x) \mid\mid \pi_{\text{ref}}(y \mid x)\right] \tag{3}$$

where $\beta$ is a parameter controlling the deviation from the base reference policy $\pi_{\text{ref}}$, namely the initial SFT model $\pi^{\text{SFT}}$. In practice, the language model policy $\pi_\theta$ is also initialized to $\pi^{\text{SFT}}$. The added constraint is important, as it prevents the model from deviating too far from the distribution on which the reward model is accurate, as well as maintaining the generation diversity and preventing mode-collapse to single high-reward answers. Due to the discrete nature of language generation, this objective is not differentiable and is typically optimized with reinforcement learning. The standard approach [48, 38, 1, 26] has been to construct the reward function $r(x, y) = r_\phi(x, y) - \beta(\log \pi_\theta(y \mid x) - \log \pi_{\text{ref}}(y \mid x))$, and maximize using PPO [37].

## 4 Direct Preference Optimization

Motivated by the challenges of applying reinforcement learning algorithms on large-scale problems such as fine-tuning language models, our goal is to derive a simple approach for policy optimization using preferences directly. Unlike prior RLHF methods, which learn a reward and then optimize it via RL, our approach bypasses the reward modeling step and directly optimizes a language model using preference data. As we will describe next in detail, our key insight is to leverage an analytical mapping from reward functions to optimal policies, which enables us to transform a loss function over reward functions into a loss function over policies. This change-of-variables approach allows us to skip the explicit reward modeling step, while still optimizing under existing models of human preferences, such as the Bradley-Terry model. In essence, the policy network represents both the language model and the reward.

**Deriving the DPO objective.** We start with the same RL objective as prior work, Eq. 3, under a general reward function $r$. Following prior work [29, 28, 17, 15], it is straightforward to show that the optimal solution to the KL-constrained reward maximization objective in Eq. 3 takes the form:

$$\pi_r(y \mid x) = \frac{1}{Z(x)} \pi_{\text{ref}}(y \mid x) \exp\left(\frac{1}{\beta} r(x, y)\right), \tag{4}$$

where $Z(x) = \sum_y \pi_{\text{ref}}(y \mid x) \exp\left(\frac{1}{\beta} r(x, y)\right)$ is the partition function. See Appendix A.1 for a complete derivation. Even if we use the MLE estimate $r_\phi$ of the ground-truth reward function $r^*$, it is still expensive to estimate the partition function $Z(x)$ [17, 15], which makes this representation hard to utilize in practice. However, we can rearrange Eq. 4 to express the reward function in terms of its corresponding optimal policy $\pi_r$, the reference policy $\pi_{\text{ref}}$, and the unknown partition function $Z(\cdot)$. Specifically, we first take the logarithm of both sides of Eq. 4 and then with some algebra we obtain:

$$r(x, y) = \beta \log \frac{\pi_r(y \mid x)}{\pi_{\text{ref}}(y \mid x)} + \beta \log Z(x). \tag{5}$$

We can apply this reparameterization to the ground-truth reward $r^*$ and corresponding optimal model $\pi^*$. Fortunately, the Bradley-Terry model depends only on the difference of rewards between two completions, i.e., $p^*(y_1 \succ y_2 \mid x) = \sigma(r^*(x, y_1) - r^*(x, y_2))$. Substituting the reparameterization in Eq. 5 for $r^*(x, y)$ into the preference model Eq. 1, the partition function cancels, and we can express the human preference probability in terms of only the optimal policy $\pi^*$ and reference policy $\pi_{\text{ref}}$. Thus, the optimal RLHF policy $\pi^*$ under the Bradley-Terry model satisfies the preference model:

$$p^*(y_1 \succ y_2 \mid x) = \frac{1}{1 + \exp\left(\beta \log \frac{\pi^*(y_2|x)}{\pi_{\text{ref}}(y_2|x)} - \beta \log \frac{\pi^*(y_1|x)}{\pi_{\text{ref}}(y_1|x)}\right)} \tag{6}$$

The derivation is in Appendix A.2. While Eq. 6 uses the Bradley-Terry model, we can similarly derive expressions under the more general Plackett-Luce models [30, 21], shown in Appendix A.3.

Now that we have the probability of human preference data in terms of the optimal policy rather than the reward model, we can formulate a maximum likelihood objective for a parametrized policy $\pi_\theta$. Analogous to the reward modeling approach (i.e. Eq. 2), our policy objective becomes:

$$\mathcal{L}_{\text{DPO}}(\pi_\theta; \pi_{\text{ref}}) = -\mathbb{E}_{(x, y_w, y_l) \sim \mathcal{D}} \left[ \log \sigma \left( \beta \log \frac{\pi_\theta(y_w \mid x)}{\pi_{\text{ref}}(y_w \mid x)} - \beta \log \frac{\pi_\theta(y_l \mid x)}{\pi_{\text{ref}}(y_l \mid x)} \right) \right]. \tag{7}$$

This way, we simultaneously bypass the explicit reward modeling step while also avoiding the need to perform reinforcement learning optimization. Moreover, since our procedure is equivalent to fitting

a reparametrized Bradley-Terry model, it enjoys certain theoretical properties, such as consistencies under suitable assumption of the preference data distribution [4]. In Section 5, we further discuss theoretical properties of DPO in relation to other works.

**What does the DPO update do?** For a mechanistic understanding of DPO, it is useful to analyze the gradient of the loss function $\mathcal{L}_{\text{DPO}}$. The gradient with respect to the parameters $\theta$ can be written as:

$$\nabla_\theta \mathcal{L}_{\text{DPO}}(\pi_\theta; \pi_{\text{ref}}) =$$
$$- \beta \mathbb{E}_{(x,y_w,y_l)\sim\mathcal{D}}\bigg[ \underbrace{\sigma(\hat{r}_\theta(x,y_l) - \hat{r}_\theta(x,y_w))}_{\text{higher weight when reward estimate is wrong}} \bigg[ \underbrace{\nabla_\theta \log \pi(y_w \mid x)}_{\text{increase likelihood of } y_w} - \underbrace{\nabla_\theta \log \pi(y_l \mid x)}_{\text{decrease likelihood of } y_l} \bigg] \bigg],$$

where $\hat{r}_\theta(x,y) = \beta \log \frac{\pi_\theta(y|x)}{\pi_{\text{ref}}(y|x)}$ is the reward implicitly defined by the language model $\pi_\theta$ and reference model $\pi_{\text{ref}}$ (more in Section 5). Intuitively, the gradient of the loss function $\mathcal{L}_{\text{DPO}}$ increases the likelihood of the preferred completions $y_w$ and decreases the likelihood of dispreferred completions $y_l$. Importantly, the examples are weighed by how much higher the implicit reward model $\hat{r}_\theta$ rates the dispreferred completions, scaled by $\beta$, i.e, how incorrectly the implicit reward model orders the completions, accounting for the strength of the KL constraint. Our experiments suggest the importance of this weighting, as a naïve version of this method without the weighting coefficient can cause the language model to degenerate (Appendix Table 3).

**DPO outline.** The general DPO pipeline is as follows: 1) Sample completions $y_1, y_2 \sim \pi_{\text{ref}}(\cdot \mid x)$ for every prompt $x$, label with human preferences to construct the offline dataset of preferences $\mathcal{D} = \{x^{(i)}, y_w^{(i)}, y_l)^{(i)}\}_{i=1}^N$ and 2) optimize the language model $\pi_\theta$ to minimize $\mathcal{L}_{\text{DPO}}$ for the given $\pi_{\text{ref}}$ and $\mathcal{D}$ and desired $\beta$. In practice, one would like to reuse preference datasets publicly available, rather than generating samples and gathering human preferences. Since the preference datasets are sampled using $\pi^{\text{SFT}}$, we initialize $\pi_{\text{ref}} = \pi^{\text{SFT}}$ whenever available. However, when $\pi^{\text{SFT}}$ is not available, we initialize $\pi_{\text{ref}}$ by maximizing likelihood of preferred completions $(x,y_w)$, that is, $\pi_{\text{ref}} = \arg\max_\pi \mathbb{E}_{x,y_w\sim\mathcal{D}} [\log \pi(y_w \mid x)]$. This procedure helps mitigate the distribution shift between the true reference distribution which is unavailable, and $\pi_{\text{ref}}$ used by DPO. Further details related to the implementation and hyperparameters can be found in Appendix B.

## 5 Theoretical Analysis of DPO

In this section, we give further interpretation of the DPO method, provide theoretical backing, and relate advantages of DPO to issues with actor critic algorithms used for RLHF (such as PPO [37]).

### 5.1 Your Language Model Is Secretly a Reward Model

DPO is able to bypass both explicit reward estimation and RL to learn the policy using a single maximum likelihood objective. However, the optimization objective Eq. 5 is equivalent to a Bradley-Terry model with a reward function $r^*(x,y) = \beta \log \frac{\pi_\theta^*(y|x)}{\pi_{\text{ref}}(y|x)}$ and we optimize our parametric model $\pi_\theta$, equivalently to the reward model optimization in Eq. 2 under the this change of variables. In this section we will build the theory behind this reparameterization, show that it does not constrain the class of learned reward models, and allows for the exact recovery of the optimal policy. We begin with by defining an equivalence relation between reward functions.

**Definition 1.** *We say that two reward functions $r(x,y)$ and $r'(x,y)$ are equivalent iff $r(x,y) - r'(x,y) = f(x)$ for some function $f$.*

It is easy to see that this is indeed an equivalence relation, which partitions the set of reward functions into classes. We can state the following two lemmas:

**Lemma 1.** *Under the Plackett-Luce, and in particular the Bradley-Terry, preference framework, two reward functions from the same class induce the same preference distribution.*

**Lemma 2.** *Two reward functions from the same equivalence class induce the same optimal policy under the constrained RL problem.*

The proofs are straightforward and we defer them to Appendix A.5. The first lemma is a well-known under-specification issue with the Plackett-Luce family of models [30]. Due to this under-specification,

we usually have to impose additional identifiability constraints to achieve any guarantees on the MLE estimates from Eq. 2 [4]. The second lemma states that all reward functions from the same class yield the same optimal policy, hence for our final objective, we are only interested in recovering an arbitrary reward function from the optimal class. We prove the following Theorem in Appendix A.6:

**Theorem 1.** *Under mild assumptions, all reward classes consistent with the Plackett-Luce (and Bradley-Terry in particular) models can be represented with the reparameterization $r(x, y) = \beta \log \frac{\pi(y|x)}{\pi_{ref}(y|x)}$ for some model $\pi(y \mid x)$ and a given reference model $\pi_{ref}(y \mid x)$.*

*Proof Sketch.* Consider any reward function $r(x, y)$, which induces a corresponding optimal model $\pi_r(y \mid x)$, specified by Eq. 4. We will show that a reward function from the equivalence class of $r$ can be represented using the reparameterization given above. We define the projection $f$ as

$$f(r; \pi_{\text{ref}}, \beta)(x, y) = r(x, y) - \beta \log \sum_y \pi_{\text{ref}}(y \mid x) \exp\left(\frac{1}{\beta} r(x, y)\right) \tag{8}$$

The operator $f$ simply normalizes the reward function with the logarithm of the partition function of $\pi_r$. Since the added normalization term is only a function of the prefix $x$, $f(r; \pi_{\text{ref}}, \beta)(x, y)$ is a reward function in the equivalence class of $r(x, y)$. Finally, replacing $r$ with the RHS of Eq. 5 (which holds for any reward function), we have $f(r; \pi_{\text{ref}}, \beta)(x, y) = \beta \log \frac{\pi_r(y|x)}{\pi_{\text{ref}}(y|x)}$. That is, the projection $f$ produces a member of the equivalence class of $r$ with the desired form, and we do not lose any generality in our reward model from the proposed reparameterization. $\square$

We can alternatively view Theorem 1 as specifying exactly which reward function within each equivalence class the DPO reparameterization selects, that is, the reward function satisfying:

$$\sum_y \underbrace{\pi_{\text{ref}}(y \mid x) \exp\left(\frac{1}{\beta} r(x, y)\right)}_{=\pi(y|x), \text{ using Thm. 1 reparam.}} = 1, \tag{9}$$

i.e., $\pi(y \mid x)$ is a valid distribution (probabilities are positive and sum to 1). However, following Eq. 4, we can see that Eq. 9 is the partition function of the optimal policy induced by the reward function $r(x, y)$. The key insight of the DPO algorithm is that we can impose certain constraints on the under-constrained Plackett-Luce (and Bradley-Terry in particular) family of preference models, such that we preserve the class of representable reward models, but explicitly make the optimal policy in Eq. 4 analytically tractable for all prompts $x$.

## 5.2 Instability of Actor-Critic Algorithms

We can also use our framework to diagnose instabilities with standard actor-critic algorithms used for the RLHF, such as PPO. We follow the RLHF pipeline and focus on the RL fine-tuning step outlined in Section 3. We can draw connections to the control as inference framework [20] for the constrained RL problem outlined in 3. We assume a parameterized model $\pi_\theta(y \mid x)$ and minimize $\mathbb{D}_{\text{KL}}[\pi_\theta(y|x) \mid\mid \pi^*(y \mid x)]$ where $\pi^*$ is the optimal policy from Eq. 7 induced by the reward function $r_\phi(y, x)$. With some algebra this leads to the optimization objective:

$$\max_{\pi_\theta} \mathbb{E}_{\pi_\theta(y|x)} \left[ \underbrace{r_\phi(x, y) - \beta \log \sum_y \pi_{\text{ref}} \exp\left(\frac{1}{\beta} r_\phi(x, y)\right)}_{f(r_\phi, \pi_{\text{ref}}, \beta)} - \underbrace{\beta \log \frac{\pi_\theta(y \mid x)}{\pi_{\text{ref}}(y \mid x)}}_{\text{KL}} \right] \tag{10}$$

This is the same objective optimized in prior works [48, 38, 1, 26] using the DPO-equivalent reward for the reward class of $r_\phi$. In this setting, we can interpret the normalization term in $f(r_\phi, \pi_{\text{ref}}, \beta)$ as the soft value function of the reference policy $\pi_{\text{ref}}$. While this term does not affect the optimal solution, without it, the policy gradient of the objective could have high variance, making learning unstable. We can accommodate for the normalization term using a learned value function, but that can also be difficult to optimize. Alternatively, prior works have normalized rewards using a human completion baseline, essentially a single sample Monte-Carlo estimate of the normalizing term. In contrast the DPO reparameterization yields a reward function that does not require any baselines.

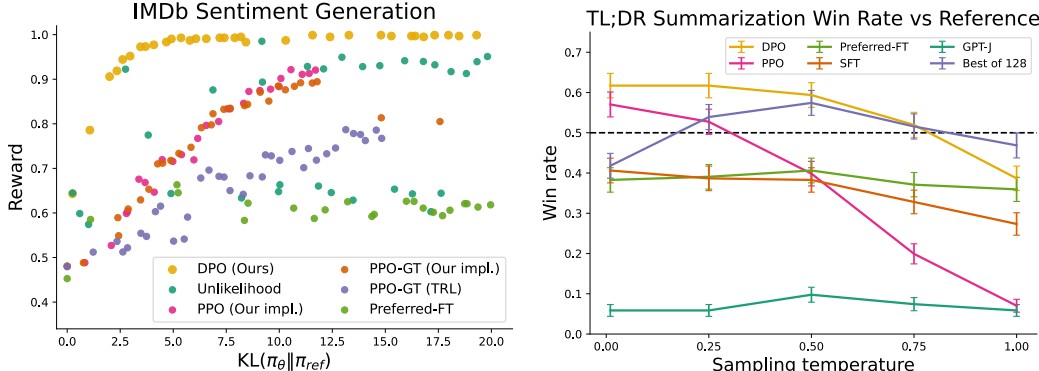

Figure 2: **Left.** The frontier of expected reward vs KL to the reference policy. DPO provides the highest expected reward for all KL values, demonstrating the quality of the optimization. **Right.** TL;DR summarization win rates vs. human-written summaries, using GPT-4 as evaluator. DPO exceeds PPO's best-case performance on summarization, while being more robust to changes in the sampling temperature.

## 6 Experiments

In this section, we empirically evaluate DPO's ability to train policies directly from preferences. First, in a well-controlled text-generation setting, we ask: how efficiently does DPO trade off maximizing reward and minimizing KL-divergence with the reference policy, compared to common preference learning algorithms such as PPO? Next, we evaluate DPO's performance on larger models and more difficult RLHF tasks, including summarization and dialogue. We find that with almost no tuning of hyperparameters, DPO tends to perform as well or better than strong baselines like RLHF with PPO as well as returning the best of $N$ sampled trajectories under a learned reward function. Before presenting these results, we describe the experimental set-up; additional details are in Appendix C.

**Tasks.** Our experiments explore three different open-ended text generation tasks. For all experiments, algorithms learn a policy from a dataset of preferences $\mathcal{D} = \left\{ x^{(i)}, y_w^{(i)}, y_l^{(i)} \right\}_{i=1}^{N}$. In **controlled sentiment generation**, $x$ is a prefix of a movie review from the IMDb dataset [22], and the policy must generate $y$ with positive sentiment. In order to perform a controlled evaluation, for this experiment we *generate* preference pairs over generations using a pre-trained sentiment classifier, where $p(\text{positive} \mid x, y_w) > p(\text{positive} \mid x, y_l)$. For SFT, we fine-tune GPT-2-large until convergence on reviews from the train split of the IMDB dataset (further details in App C.1). In **summarization**, $x$ is a forum post from Reddit; the policy must generate a summary $y$ of the main points in the post. Following prior work, we use the Reddit TL;DR summarization dataset [41] along with human preferences gathered by Stiennon et al.. We use an SFT model fine-tuned on human-written forum post summaries[2] with the TRLX [42] framework for RLHF. The human preference dataset was gathered by Stiennon et al. on samples from a different, but similarly-trained, SFT model. Finally, in **single-turn dialogue**, $x$ is a human query, which may be anything from a question about astrophysics to a request for relationship advice. A policy must produce an engaging and helpful response $y$ to a user's query; we use the Anthropic Helpful and Harmless dialogue dataset [1], containing 170k dialogues between a human and an automated assistant. Each transcript ends with a pair of responses generated by a large (although unknown) language model along with a preference label denoting the human-preferred response. In this setting, no pre-trained SFT model is available; we therefore fine-tune an off-the-shelf language model on only the preferred completions to form the SFT model.

**Evaluation.** Our experiments use two different approaches to evaluation. In order to analyze the effectiveness of each algorithm in optimizing the constrained reward maximization objective, in the controlled sentiment generation setting we evaluate each algorithm by its frontier of achieved reward and KL-divergence from the reference policy; this frontier is computable because we have acccess to the ground-truth reward function (a sentiment classifier). However, in the real world, the ground truth reward function is not known; therefore, we evaluate algorithms with their *win rate* against a baseline policy, using GPT-4 as a proxy for human evaluation of summary quality and response helpfulness in the summarization and single-turn dialogue settings, respectively. For summarization, we use reference summaries in the test set as the baseline; for dialogue, we use the preferred response in the

---

[2] https://huggingface.co/CarperAI/openai_summarize_tldr_sft

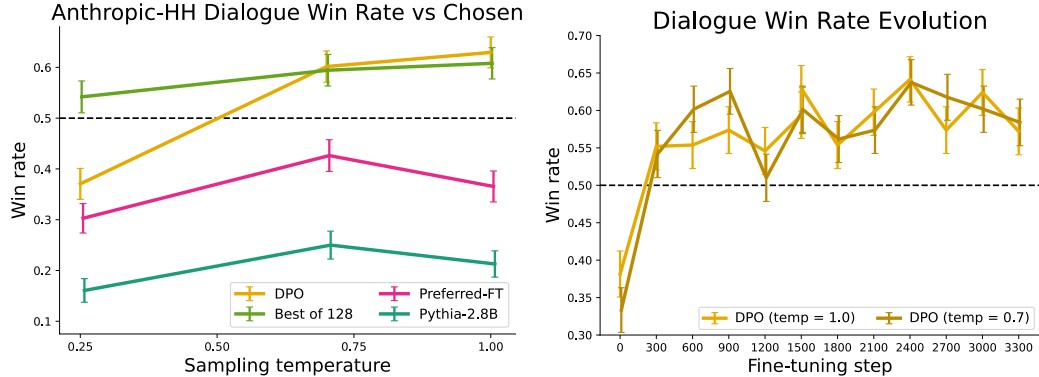

Figure 3: **Left.** Win rates computed by GPT-4 for Anthropic-HH one-step dialogue; DPO is the only method that improves over chosen summaries in the Anthropic-HH test set. **Right.** Win rates for different sampling temperatures over the course of training. DPO's improvement over the dataset labels is fairly stable over the course of training for different sampling temperatures.

test dataset as the baseline. While existing studies suggest LMs can be better automated evaluators than existing metrics [10], we conduct a human study to justify our usage of GPT-4 for evaluation in Sec. 6.4. We find GPT-4 judgments correlate strongly with humans, with human agreement with GPT-4 typically similar or higher than inter-human annotator agreement.

**Methods.** In addition to DPO, we evaluate several existing approaches to training language models to adhere to human preferences. Most simply, we explore zero-shot prompting with **GPT-J** [43] in the summarization task and 2-shot prompting with **Pythia-2.8B** [3] in the dialogue task. In addition, we evaluate the **SFT** model as well as **Preferred-FT**, which is a model fine-tuned with supervised learning on the chosen completion $y_w$ from either the SFT model (in controlled sentiment and summarization) or a generic LM (in single-turn dialogue). Another pseudo-supervised method is **Unlikelihood**, which simply optimizes the policy to maximize the probability assigned to $y_w$ and *minimize* the probability assigned to $y_l$; we use an optional coefficient $\alpha \in [0, 1]$ on the 'unlikelihood' term. We also consider **PPO** [37] using a reward function learned from the preference data and **PPO-GT**, which is an oracle that learns from the ground truth reward function available in the controlled sentiment setting. In our sentiment experiments, we use two implementations of PPO-GT, one of-the-shelf version [42] as well as a modified version that normalizes rewards and further tunes hyperparameters to improve performance (we also use these modifications when running 'normal' PPO with learned rewards). Finally, we consider the **Best of** $N$ baseline, sampling $N$ responses from the SFT model (or Preferred-FT in dialogue) and returning the highest-scoring response according to a reward function learned from the preference dataset. This high-performing method decouples the quality of the reward model from the PPO optimization, but is computationally impractical even for moderate $N$ as it requires sampling $N$ completions for every query at test time.

### 6.1 How well can DPO optimize the RLHF objective?

The KL-constrained reward maximization objective used in typical RLHF algorithms balances exploitation of reward while restricting the policy from deviating far from the reference policy. Therefore, when comparing algorithms, we must take into account both reward achieved as well as the KL discrepancy; achieving slightly higher reward but with much higher KL is not necessarily desirable. Figure 2 shows the reward-KL frontier for various algorithms in the sentiment setting. We execute multiple training runs for each algorithm, using a different hyperparameter for policy conservativeness in each run (target KL $\in \{3, 6, 9, 12\}$ for PPO, $\beta \in \{0.05, 0.1, 1, 5\}$, $\alpha \in \{0.05, 0.1, 0.5, 1\}$ for unlikelihood, random seeds for preferred-FT). This sweep includes 22 runs in total. After each 100 training steps until convergence, we evaluate each policy on a set of test prompts, computing the average reward under the true reward function as well as the average sequence-level KL[3] with the reference policy $\mathrm{KL}\left(\pi \| \pi_{\mathrm{ref}}\right)$. We find that DPO produces by far the most efficient frontier, achieving the highest reward while still achieving low KL. This result is particularly notable for multiple reasons. First, DPO and PPO optimize the same objective, but DPO is notably more efficient;

---

[3]That is, the sum of the per-timestep KL-divergences.

DPO's reward/KL tradeoff strictly dominates PPO. Second, DPO achieves a better frontier than PPO, *even when PPO can access ground truth rewards* (PPO-GT).

## 6.2 Can DPO scale to real preference datasets?

Next, we evaluate fine-tuning performance of DPO on summarization and single-turn dialogue. For summarization, automatic evaluation metrics such as ROUGE can be poorly correlated with human preferences [38], and prior work has found that fine-tuning LMs using PPO on human preferences to provide more effective summaries. We evaluate different methods by sampling completions on the test split of TL;DR summarization dataset, and computing the average win rate against reference completions in the test set. The completions for all methods are sampled at temperatures varying from 0.0 to 1.0, and the win rates are shown in Figure 2 (right). DPO, PPO and Preferred-FT all fine-tune the same GPT-J SFT model[4]. We find that DPO has a win rate of approximately 61% at a temperature of 0.0, exceeding the performance of PPO at 57% at its optimal sampling temperature of 0.0. DPO also achieves a higher maximum win rate compared to the best of $N$ baseline. We note that we did not meaningfully tune DPO's $\beta$ hyperparameter, so these results may underestimate DPO's potential. Moreover, we find DPO to be much more robust to the sampling temperature than PPO, the performance of which can degrade to that of the base GPT-J model at high temperatures. Preferred-FT does not improve significantly over the SFT model. We also compare DPO and PPO head-to-head in human evaluations in Section 6.4, where DPO samples at temperature 0.25 were preferred 58% times over PPO samples at temperature 0.

On single-turn dialogue, we evaluate the different methods on the subset of the test split of the Anthropic HH dataset [1] with one step of human-assistant interaction. GPT-4 evaluations use the preferred completions on the test as the reference to compute the win rate for different methods. As there is no standard SFT model for this task, we start with a pre-trained Pythia-2.8B, use Preferred-FT to train a reference model on the chosen completions such that completions are within distribution of the model, and then train using DPO. We also compare against the best of 128 Preferred-FT completions (we found the Best of $N$ baseline plateaus at 128 completions for this task; see Appendix Figure 4) and a 2-shot prompted version of the Pythia-2.8B base model, finding DPO performs as well or better for the best-performing temperatures for each method. We also evaluate an RLHF model trained with PPO on the Anthropic HH dataset [5] from a well-known source [6], but are unable to find a prompt or sampling temperature that gives performance better than the base Pythia-2.8B model. Based on our results from TL;DR and the fact that both methods optimize the same reward function, we consider Best of 128 a rough proxy for PPO-level performance. Overall, DPO is the only computationally efficient method that improves over the preferred completions in the Anthropic HH dataset, and provides similar or better performance to the computationally demanding Best of 128 baseline. Finally, Figure 3 shows that DPO converges to its best performance relatively quickly.

## 6.3 Generalization to a new input distribution

To further compare the performance of PPO and DPO under distribution shifts, we evaluate the PPO and DPO policies from our Reddit TL;DR summarization experiment on a different distribution, news articles in the test split of the CNN/DailyMail dataset [24], using the best sampling temperatures from TL;DR (0 and 0.25). The results are presented in Table 1. We computed the GPT-4 win rate against the ground-truth summaries in the datasets, using the same GPT-4 (C) prompt we used for Reddit TL;DR, but replacing the words "forum post" with "news article".

|  | Win rate vs. ground truth | |
|---|---|---|
| **Alg.** | Temp 0 | Temp 0.25 |
| DPO | 0.36 | 0.31 |
| PPO | 0.26 | 0.23 |

Table 1: GPT-4 win rates vs. ground truth summaries for out-of-distribution CNN/DailyMail input articles.

For this new distribution, DPO continues to outperform the PPO policy by a significant margin. This experiment provides initial evidence that DPO policies can generalize similarly well to PPO policies, even though DPO does not use the additional unlabeled Reddit TL;DR prompts that PPO uses.

---

[4] https://huggingface.co/CarperAI/openai_summarize_tldr_sft
[5] https://huggingface.co/reciprocate/ppo_hh_pythia-6B
[6] https://github.com/CarperAI/trlx/tree/main/examples/hh

### 6.4 Validating GPT-4 judgments with human judgments

We conduct a human study to verify the reliability of GPT-4's judgments, using the results of the TL;DR summarization experiment and two different GPT-4 prompts. The **GPT-4 (S)** (simple) prompt simply asks for which summary better-summarizes the important information in the post. The **GPT-4 (C)** (concise) prompt also asks for which summary is more concise; we evaluate this prompt because we find that GPT-4 prefers longer, more repetitive summaries than humans do with the **GPT-4 (S)** prompt. See Appendix C.2 for the complete prompts. We perform three comparisons, using the highest (DPO, temp. 0.25), the lowest (PPO, temp. 1.0), and a

middle-performing (SFT, temp. 0.25) method with the aim of covering a diversity of sample qualities; all three methods are compared against greedily-sampled PPO (its best-performing temperature). We find that with both prompts, GPT-4 tends to agree with humans about as often as humans agree with each other, suggesting that GPT-4 is a reasonable proxy for human evaluations (due to limited human raters, we only collect multiple human judgments for the DPO and PPO-1 comparisons). Overall, the **GPT-4 (C)** prompt generally provides win rates more representative of humans; we therefore use this prompt for the main results in Section 6.2. For additional details about the human study, including the web interface presented to raters and the list of human volunteers, see Appendix D.3.

|  | DPO | SFT | PPO-1 |
|---|---|---|---|
| N respondents | 272 | 122 | 199 |
| GPT-4 (S) win % | 47 | 27 | 13 |
| GPT-4 (C) win % | 54 | 32 | 12 |
| Human win % | 58 | 43 | 17 |
| GPT-4 (S)-H agree | 70 | 77 | 86 |
| GPT-4 (C)-H agree | 67 | 79 | 85 |
| H-H agree | 65 | - | 87 |

Table 2: Comparing human and GPT-4 win rates and per-judgment agreement on TL;DR summarization samples. **Humans agree with GPT-4 about as much as they agree with each other.** Each experiment compares a summary from the stated method with a summary from PPO with temperature 0.

## 7 Discussion

Learning from preferences is a powerful, scalable framework for training capable, aligned language models. We have introduced DPO, a simple training paradigm for training language models from preferences without reinforcement learning. Rather than coercing the preference learning problem into a standard RL setting in order to use off-the-shelf RL algorithms, DPO identifies a mapping between language model policies and reward functions that enables training a language model to satisfy human preferences *directly*, with a simple cross-entropy loss, without reinforcement learning or loss of generality. With virtually no tuning of hyperparameters, DPO performs similarly or better than existing RLHF algorithms, including those based on PPO; DPO thus meaningfully reduces the barrier to training more language models from human preferences.

**Limitations & Future Work.** Our results raise several important questions for future work. How does the DPO policy generalize out of distribution, compared with learning from an explicit reward function? Our initial results suggest that DPO policies can generalize similarly to PPO-based models, but more comprehensive study is needed. For example, can training with self-labeling from the DPO policy similarly make effective use of unlabeled prompts? On another front, how does reward over-optimization manifest in the direct preference optimization setting, and is the slight decrease in performance in Figure 3-right an instance of it? Additionally, while we evaluate models up to 6B parameters, exploration of scaling DPO to state-of-the-art models orders of magnitude larger is an exciting direction for future work. Regarding evaluations, we find that the win rates computed by GPT-4 are impacted by the prompt; future work may study the best way to elicit high-quality judgments from automated systems. Finally, many possible applications of DPO exist beyond training language models from human preferences, including training generative models in other modalities.

## Acknowledgements

EM gratefully acknowledges funding from a Knight-Hennessy Graduate Fellowship. CF and CM are CIFAR Fellows. This work was supported in part by the Stanford Accelerator for Learning (SAL) and Stanford Institute for Human-Centered Artificial Intelligence (HAI) *Generative AI for the Future of Learning* seed grant program. The Stanford Center for Research on Foundation Models (CRFM) provided part of the compute resources used for the experiments in this work. This work was supported in part by ONR grant N00014-20-1-2675.

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
