## Author Contributions

**All authors** provided valuable contributions to designing, analyzing, and iterating on experiments, writing and editing the paper, and generally managing the project's progress.

**RR** proposed using autoregressive reward models in discussions with **EM**; derived the DPO objective; proved the theoretical properties of the algorithm and wrote the relevant sections and appendices. He also suggested and helped with organizing experiments and contributed some of the PPO and reward learning baselines.

**AS** initiated the discussion on using weighted regression methods as an alternative to PPO; initiated project-related organization, wrote initial analysis connecting DPO with weighted regression and unlikelihood; design and iterations of DPO + baseline implementations, initial exploratory experiments for DPO; substantial experiment organization and design (datasets, baselines, evaluation); led model training and evaluation for controlled sentiment generation and summarization; design iterations for GPT-4 evaluation (particularly summarization); substantial writing contributions to abstract, prelims/method and experiments; editing contributions to other sections.

**EM** provided input on early discussions on learning autoregressive reward functions; wrote the first implementation of DPO and ran the first DPO experiments; trained the large-scale (summarization and dialogue) DPO models used in paper experiments; conducted initial GPT-4 win rate evaluations and set up related infrastructure; recruited participants for, conducted, and analyzed results from the human study; wrote the abstract, introduction, related work, discussion, and most of experiments; and assisted with editing the rest of the paper.

**CF, CM, & SE** supervised the research, suggested ideas and experiments, and assisted in writing the paper.

## A   Mathematical Derivations

### A.1   Deriving the Optimum of the KL-Constrained Reward Maximization Objective

In this appendix, we will derive Eq. 4. Analogously to Eq. 3, we optimize the following objective:

$$\max_{\pi} \mathbb{E}_{x \sim \mathcal{D}, y \sim \pi} \big[ r(x, y) \big] - \beta \mathbb{D}_{\text{KL}} \big[ \pi(y|x) || \pi_{\text{ref}}(y|x) \big] \tag{11}$$

under any reward function $r(x, y)$, reference model $\pi_{\text{ref}}$ and a general non-parametric policy class. We now have:

$$\max_{\pi} \mathbb{E}_{x \sim \mathcal{D}, y \sim \pi} \big[ r(x, y) \big] - \beta \mathbb{D}_{\text{KL}} \big[ \pi(y|x) \, || \, \pi_{\text{ref}}(y|x) \big]$$

$$= \max_{\pi} \mathbb{E}_{x \sim \mathcal{D}} \mathbb{E}_{y \sim \pi(y|x)} \left[ r(x, y) - \beta \log \frac{\pi(y|x)}{\pi_{\text{ref}}(y|x)} \right]$$

$$= \min_{\pi} \mathbb{E}_{x \sim \mathcal{D}} \mathbb{E}_{y \sim \pi(y|x)} \left[ \log \frac{\pi(y|x)}{\pi_{\text{ref}}(y|x)} - \frac{1}{\beta} r(x, y) \right]$$

$$= \min_{\pi} \mathbb{E}_{x \sim \mathcal{D}} \mathbb{E}_{y \sim \pi(y|x)} \left[ \log \frac{\pi(y|x)}{\frac{1}{Z(x)} \pi_{\text{ref}}(y|x) \exp \left( \frac{1}{\beta} r(x, y) \right)} - \log Z(x) \right] \tag{12}$$

where we have partition function:

$$Z(x) = \sum_{y} \pi_{\text{ref}}(y|x) \exp \left( \frac{1}{\beta} r(x, y) \right).$$

Note that the partition function is a function of only $x$ and the reference policy $\pi_{\text{ref}}$, but does not depend on the policy $\pi$. We can now define

$$\pi^*(y|x) = \frac{1}{Z(x)} \pi_{\text{ref}}(y|x) \exp \left( \frac{1}{\beta} r(x, y) \right),$$

which is a valid probability distribution as $\pi^*(y|x) \geq 0$ for all $y$ and $\sum_y \pi^*(y|x) = 1$. Since $Z(x)$ is not a function of $y$, we can then re-organize the final objective in Eq 12 as:

$$\min_\pi \mathbb{E}_{x \sim \mathcal{D}} \left[ \mathbb{E}_{y \sim \pi(y|x)} \left[ \log \frac{\pi(y|x)}{\pi^*(y|x)} \right] - \log Z(x) \right] = \tag{13}$$

$$\min_\pi \mathbb{E}_{x \sim \mathcal{D}} \left[ \mathbb{D}_{\text{KL}}(\pi(y|x) \;||\; \pi^*(y|x)) + Z(x) \right] \tag{14}$$

Now, since $Z(x)$ does not depend on $\pi$, the minimum is achieved by the policy that minimizes the first KL term. Gibbs' inequality tells us that the KL-divergence is minimized at 0 if and only if the two distributions are identical. Hence we have the optimal solution:

$$\pi(y|x) = \pi^*(y|x) = \frac{1}{Z(x)} \pi_{\text{ref}}(y|x) \exp \left( \frac{1}{\beta} r(x, y) \right) \tag{15}$$

for all $x \in \mathcal{D}$. This completes the derivation.

### A.2 Deriving the DPO Objective Under the Bradley-Terry Model

It is straightforward to derive the DPO objective under the Bradley-Terry preference model as we have

$$p^*(y_1 \succ y_2|x) = \frac{\exp\left(r^*(x, y_1)\right)}{\exp\left(r^*(x, y_1)\right) + \exp\left(r^*(x, y_2)\right)} \tag{16}$$

In Section 4 we showed that we can express the (unavailable) ground-truth reward through its corresponding optimal policy:

$$r^*(x, y) = \beta \log \frac{\pi^*(y|x)}{\pi_{\text{ref}}(y|x)} + \beta \log Z(x) \tag{17}$$

Substituting Eq. 17 into Eq. 16 we obtain:

$$p^*(y_1 \succ y_2|x) = \frac{\exp\left(\beta \log \frac{\pi^*(y_1|x)}{\pi_{\text{ref}}(y_1|x)} + \beta \log Z(x)\right)}{\exp\left(\beta \log \frac{\pi^*(y_1|x)}{\pi_{\text{ref}}(y_1|x)} + \beta \log Z(x)\right) + \exp\left(\beta \log \frac{\pi^*(y_2|x)}{\pi_{\text{ref}}(y_2|x)} + \beta \log Z(x)\right)}$$

$$= \frac{1}{1 + \exp\left(\beta \log \frac{\pi^*(y_2|x)}{\pi_{\text{ref}}(y_2|x)} - \beta \log \frac{\pi^*(y_1|x)}{\pi_{\text{ref}}(y_1|x)}\right)}$$

$$= \sigma \left( \beta \log \frac{\pi^*(y_1|x)}{\pi_{\text{ref}}(y_1|x)} - \beta \log \frac{\pi^*(y_2|x)}{\pi_{\text{ref}}(y_2|x)} \right).$$

The last line is the per-instance loss in Equation 7.

### A.3 Deriving the DPO Objective Under the Plackett-Luce Model

The Plackett-Luce model [30, 21] is a generalization of the Bradley-Terry model over rankings (rather than just pair-wise comparisons). Similar to to the Bradley-Terry model, it stipulates that when presented with a set of possible choices, people prefer a choice with probability proportional to the value of some latent reward function for that choice. In our context, when presented with a prompt $x$ and a set of $K$ answers $y_1, \ldots, y_K$ a user would output a permutation $\tau : [K] \to [K]$, giving their ranking of the answers. The Plackett-Luce model stipulates that

$$p^*(\tau|y_1, \ldots, y_K, x) = \prod_{k=1}^{K} \frac{\exp(r^*(x, y_{\tau(k)}))}{\sum_{j=k}^{K} \exp(r^*(x, y_{\tau(j)}))} \tag{18}$$

Notice that when $K = 2$, Equation 18 reduces to the Bradley-Terry model. However, for the general Plackett-Luce model, we can still utilize the results of Eq. 5 and substitute the reward function parameterized by its optimal policy. Similarly to Appendix A.2, the normalization constant $Z(x)$ cancels out and we're left with:

$$p^*(\tau|y_1, \ldots, y_K, x) = \prod_{k=1}^{K} \frac{\exp\left(\beta \log \frac{\pi^*(y_{\tau(k)}|x)}{\pi_{\text{ref}}(y_{\tau(k)}|x)}\right)}{\sum_{j=k}^{K} \exp\left(\beta \log \frac{\pi^*(y_{\tau(j)}|x)}{\pi_{\text{ref}}(y_{\tau(j)}|x)}\right)} \tag{19}$$

Similarly to the approach of Section 4, if we have access to a dataset $\mathcal{D} = \{\tau^{(i)}, y_1^{(i)}, \ldots, y_K^{(i)}, x^{(i)}\}_{i=1}^N$ of prompts and user-specified rankings, we can use a parameterized model and optimize this objective with maximum-likelihood.:

$$\mathcal{L}_{\text{DPO}}(\pi_\theta, \pi_{\text{ref}}) = -\mathbb{E}_{\tau, y_1, \ldots, y_K, x \sim \mathcal{D}} \left[ \log \prod_{k=1}^K \frac{\exp\left(\beta \log \frac{\pi_\theta(y_{\tau(k)}|x)}{\pi_{\text{ref}}(y_{\tau(k)}|x)}\right)}{\sum_{j=k}^K \exp\left(\beta \log \frac{\pi_\theta(y_{\tau(j)}|x)}{\pi_{\text{ref}}(y_{\tau(j)}|x)}\right)} \right] \tag{20}$$

### A.4 Deriving the Gradient of the DPO Objective

In this section we derive the gradient of the DPO objective:

$$\nabla_\theta \mathcal{L}_{\text{DPO}}(\pi_\theta; \pi_{\text{ref}}) = -\nabla_\theta \mathbb{E}_{(x, y_w, y_l) \sim \mathcal{D}} \left[ \log \sigma \left( \beta \log \frac{\pi_\theta(y_l|x)}{\pi_{\text{ref}}(y_l|x)} - \beta \log \frac{\pi_\theta(y_w|x)}{\pi_{\text{ref}}(y_w|x)} \right) \right] \tag{21}$$

We can rewrite the RHS of Equation 21 as

$$\nabla_\theta \mathcal{L}_{\text{DPO}}(\pi_\theta; \pi_{\text{ref}}) = -\mathbb{E}_{(x, y_w, y_l) \sim \mathcal{D}} \left[ \frac{\sigma'(u)}{\sigma(u)} \nabla_\theta(u) \right], \tag{22}$$

where $u = \beta \log \frac{\pi_\theta(y_l|x)}{\pi_{\text{ref}}(y_l|x)} - \beta \log \frac{\pi_\theta(y_w|x)}{\pi_{\text{ref}}(y_w|x)}$.

Using the properties of sigmoid function $\sigma'(x) = \sigma(x)(1 - \sigma(x))$ and $\sigma(-x) = 1 - \sigma(x)$, we obtain the final gradient

$$\nabla_\theta \mathcal{L}_{\text{DPO}}(\pi_\theta; \pi_{\text{ref}}) =$$

$$-\mathbb{E}_{(x, y_w, y_l) \sim \mathcal{D}} \left[ \beta \sigma \left( \beta \log \frac{\pi_\theta(y_w|x)}{\pi_{\text{ref}}(y_w|x)} - \beta \log \frac{\pi_\theta(y_l|x)}{\pi_{\text{ref}}(y_l|x)} \right) \left[ \nabla_\theta \log \pi(y_w \mid x) - \nabla_\theta \log \pi(y_l \mid x) \right] \right],$$

After using the reward substitution of $\hat{r}_\theta(x, y) = \beta \log \frac{\pi_\theta(y|x)}{\pi_{\text{ref}}(y|x)}$ we obtain the final form of the gradient from Section 4.

### A.5 Proof of Lemma 1 and 2

In this section, we will prove the two lemmas from Section 5.

**Lemma 1 Restated.** *Under the Plackett-Luce preference framework, and in particular the Bradley-Terry framework, two reward functions from the same equivalence class induce the same preference distribution.*

*Proof.* We say that two reward functions $r(x, y)$ and $r'(x, y)$ are from the same equivalence class if $r'(x, y) = r(x, y) + f(x)$ for some function $f$. We consider the general Plackett-Luce (with the Bradley-Terry model a special case for $K = 2$) and denote the probability distribution over rankings induced by a particular reward function $r(x, y)$ as $p_r$. For any prompt $x$, answers $y_1, \ldots, y_K$ and ranking $\tau$ we have:

$$\begin{aligned} p_{r'}(\tau | y_1, \ldots, y_K, x) &= \prod_{k=1}^K \frac{\exp(r'(x, y_{\tau(k)}))}{\sum_{j=k}^K \exp(r'(x, y_{\tau(j)}))} \\ &= \prod_{k=1}^K \frac{\exp(r(x, y_{\tau(k)}) + f(x))}{\sum_{j=k}^K \exp(r(x, y_{\tau(j)}) + f(x))} \\ &= \prod_{k=1}^K \frac{\exp(f(x)) \exp(r(x, y_{\tau(k)}))}{\exp(f(x)) \sum_{j=k}^K \exp(r(x, y_{\tau(j)}))} \\ &= \prod_{k=1}^K \frac{\exp(r(x, y_{\tau(k)}))}{\sum_{j=k}^K \exp(r(x, y_{\tau(j)}))} \\ &= p_r(\tau | y_1, \ldots, y_K, x), \end{aligned}$$

which completes the proof. $\qquad\square$

**Lemma 2 Restated.** *Two reward functions from the same equivalence class induce the same optimal policy under the constrained RL problem.*

*Proof.* Let us consider two reward functions from the same class, such that $r'(x, y) = r(x, y) + f(x)$ and, let us denote as $\pi_r$ and $\pi_{r'}$ the corresponding optimal policies. By Eq. 4, for all $x, y$ we have

$$\pi_{r'}(y|x) = \frac{1}{\sum_y \pi_{\text{ref}}(y|x) \exp\left(\frac{1}{\beta} r'(x, y)\right)} \pi_{\text{ref}}(y|x) \exp\left(\frac{1}{\beta} r'(x, y)\right)$$

$$= \frac{1}{\sum_y \pi_{\text{ref}}(y|x) \exp\left(\frac{1}{\beta} (r(x, y) + f(x))\right)} \pi_{\text{ref}}(y|x) \exp\left(\frac{1}{\beta} (r(x, y) + f(x))\right)$$

$$= \frac{1}{\exp\left(\frac{1}{\beta} f(x)\right) \sum_y \pi_{\text{ref}}(y|x) \exp\left(\frac{1}{\beta} r(x, y)\right)} \pi_{\text{ref}}(y|x) \exp\left(\frac{1}{\beta} r(x, y)\right) \exp\left(\frac{1}{\beta} f(x)\right)$$

$$= \frac{1}{\sum_y \pi_{\text{ref}}(y|x) \exp\left(\frac{1}{\beta} r(x, y)\right)} \pi_{\text{ref}}(y|x) \exp\left(\frac{1}{\beta} r(x, y)\right)$$

$$= \pi_r(y|x),$$

which completes the proof. $\qquad\square$

## A.6   Proof of Theorem 1

In this section, we will expand on the results of Theorem 1.

**Theorem 1 Restated.** *Assume, we have a reference model, such that $\pi_{ref}(y|x) > 0$ for all pairs of prompts $x$ and answers $y$ and a parameter $\beta > 0$. All reward equivalence classes, as defined in Section 5 can be represented with the reparameterization $r(x, y) = \beta \log \frac{\pi(y|x)}{\pi_{ref}(y|x)}$ for some model $\pi(y|x)$.*

*Proof.* Consider any reward function $r(x, y)$, which induces an optimal model $\pi_r(y|x)$ under the KL-constrained RL problem, with solution given by 4. Following Eq. 5, when we log-linearize both sides we obtain:

$$r(x, y) = \beta \log \frac{\pi_r(y|x)}{\pi_{\text{ref}}(y|x)} + \beta \log Z(x)$$

where $Z(x) = \sum_y \pi_{\text{ref}}(y|x) \exp\left(\frac{1}{\beta} r(x, y)\right)$ (notice that $Z(x)$ also depends on the reward function $r$). Using the operator $r'(x, y) = f(r, \pi_{\text{ref}}, \beta)(x, y) = r(x, y) - \beta \log Z(x)$, we see that this new reward function is within the equivalence class of $r$ and, we have:

$$r'(x, y) = \beta \log \frac{\pi_r(y|x)}{\pi_{\text{ref}}(y|x)}$$

which completes the proof. $\qquad\square$

We can further expand on these results. We can see that if $r$ and $r'$ are two reward functions in the same class, then

$$f(r, \pi_{\text{ref}}, \beta)(x, y) = \beta \log \frac{\pi_r(y|x)}{\pi_{\text{ref}}(y|x)} = \beta \log \frac{\pi_r'(y|x)}{\pi_{\text{ref}}(y|x)} = f(r', \pi_{\text{ref}}, \beta)(x, y)$$

where the second equality follows from Lemma 2. We have proven that the operator $f$ maps all reward functions from a particular equivalence class to the same reward function. Next, we show that for every equivalence class of reward functions, the reward function that has the reparameterization outlined in Theorem 1 is unique.

**Proposition 1.** *Assume, we have a reference model, such that $\pi_{ref}(y|x) > 0$ for all pairs of prompts $x$ and answers $y$ and a parameter $\beta > 0$. Then every equivalence class of reward functions, as defined in Section 5, has a unique reward function $r(x, y)$, which can be reparameterized as $r(x, y) = \beta \log \frac{\pi(y|x)}{\pi_{ref}(y|x)}$ for some model $\pi(y|x)$.*

*Proof.* We will proceed using proof by contradiction. Assume we have two reward functions from the same class, such that $r'(x, y) = r(x, y) + f(x)$. Moreover, assume that $r'(x, y) = \beta \log \frac{\pi'(y|x)}{\pi_{\text{ref}}(y|x)}$ for some model $\pi'(y|x)$ and $r(x, y) = \beta \log \frac{\pi(y|x)}{\pi_{\text{ref}}(y|x)}$ for some model $\pi(y|x)$, such that $\pi \neq \pi'$. We then have

$$r'(x, y) = r(x, y) + f(x) = \beta \log \frac{\pi(y|x)}{\pi_{\text{ref}}(y|x)} + f(x) = \beta \log \frac{\pi(y|x) \exp(\frac{1}{\beta} f(x))}{\pi_{\text{ref}}(y|x)} = \beta \log \frac{\pi'(y|x)}{\pi_{\text{ref}}(y|x)}$$

for all prompts $x$ and completions $y$. Then we must have $\pi(y|x) \exp(\frac{1}{\beta} f(x)) = \pi'(y|x)$. Since these are distributions, summing over $y$ on both sides, we obtain that $\exp(\frac{1}{\beta} f(x)) = 1$ and since $\beta > 0$, we must have $f(x) = 0$ for all $x$. Therefore $r(x, y) = r'(x, y)$. This completes the proof. $\square$

We have now shown that every reward class has a unique reward function that can be represented as outlined in Theorem 1, which is given by $f(r, \pi_{\text{ref}}, \beta)$ for any reward function in that class.

## B   DPO Implementation Details and Hyperparameters

DPO is relatively straightforward to implement; PyTorch code for the DPO loss is provided below:

```
import torch.nn.functional as F

def dpo_loss(pi_logps, ref_logps, yw_idxs, yl_idxs, beta):
    """
    pi_logps: policy logprobs, shape (B,)
    ref_logps: reference model logprobs, shape (B,)
    yw_idxs: preferred completion indices in [0, B-1], shape (T,)
    yl_idxs: dispreferred completion indices in [0, B-1], shape (T,)
    beta: temperature controlling strength of KL penalty

    Each pair of (yw_idxs[i], yl_idxs[i]) represents the
      indices of a single preference pair.
    """

    pi_yw_logps,  pi_yl_logps =  pi_logps[yw_idxs],  pi_logps[yl_idxs]
    ref_yw_logps, ref_yl_logps = ref_logps[yw_idxs], ref_logps[yl_idxs]

    pi_logratios  = pi_yw_logps - pi_yl_logps
    ref_logratios = ref_yw_logps - ref_yl_logps

    losses = -F.logsigmoid(beta * (pi_logratios - ref_logratios))
    rewards = beta * (pi_logps - ref_logps).detach()

    return losses, rewards
```

Unless noted otherwise, we use a $\beta = 0.1$, batch size of 64 and the Adam optimizer with a learning rate of 1e-6 by default. We linearly warmup the learning rate from 0 to 1e-6 over 150 steps. For TL;DR summarization, we use $\beta = 0.5$, while rest of the parameters remain the same.

## C   Further Details on the Experimental Set-Up

In this section, we include additional details relevant to our experimental design.

### C.1   IMDb Sentiment Experiment and Baseline Details

The prompts are prefixes from the IMDB dataset of length 2-8 tokens. We use the pre-trained sentiment classifier siebert/sentiment-roberta-large-english as a ground-truth reward model and gpt2-large as a base model. We use these larger models as we found the default ones to generate low-quality text and rewards to be somewhat inaccurate. We first use supervised fine-tuning on a subset of the IMDB data for 1 epoch. We then use this model to sample 4 completions for 25000 prefixes and create 6 preference pairs for each prefix using the ground-truth reward model. The RLHF reward model is initialized from the gpt2-large model and trained for 3 epochs on the preference datasets, and we take the checkpoint with the highest validation set accuracy. The "TRL" run uses the hyper-parameters in the TRL library. Our implementation uses larger batch samples of 1024 per PPO step.

### C.2   GPT-4 prompts for computing summarization and dialogue win rates

A key component of our experimental setup is GPT-4 win rate judgments. In this section, we include the prompts used to generate win rates for the summarization and dialogue experiments. We use gpt-4-0314 for all our experiments. The order of summaries or responses are randomly chosen for every evaluation.

**Summarization GPT-4 win rate prompt (S).**

```
Which of the following summaries does a better job of summarizing the most \
important points in the given forum post?

Post:
```

```
<post>

Summary A:
<Summary A>

Summary B:
<Summary B>

FIRST provide a one-sentence comparison of the two summaries, explaining which \
you prefer and why. SECOND, on a new line, state only "A" or "B" to indicate your \
choice. Your response should use the format:
Comparison: <one-sentence comparison and explanation>
Preferred: <"A" or "B">
```

**Summarization GPT-4 win rate prompt (C).**

```
Which of the following summaries does a better job of summarizing the most \
important points in the given forum post, without including unimportant or \
irrelevant details? A good summary is both precise and concise.

Post:
<post>

Summary A:
<Summary A>

Summary B:
<Summary B>

FIRST provide a one-sentence comparison of the two summaries, explaining which \
you prefer and why. SECOND, on a new line, state only "A" or "B" to indicate your \
choice. Your response should use the format:
Comparison: <one-sentence comparison and explanation>
Preferred: <"A" or "B">
```

**Dialogue GPT-4 win rate prompt.**

```
For the following query to a chatbot, which response is more helpful?

Query: <the user query>

Response A:
<either the test method or baseline>

Response B:
<the other response>

FIRST provide a one-sentence comparison of the two responses and explain \
which you feel is more helpful. SECOND, on a new line, state only "A" or \
"B" to indicate which response is more helpful. Your response should use \
the format:
Comparison: <one-sentence comparison and explanation>
More helpful: <"A" or "B">
```

### C.3  Unlikelihood baseline

While we include the unlikelihood baseline (simply maximizing $\log p(y_w|x)$, the log probability of the preferred response, while minimizing $\log p(y_l|x)$, the log probability of the dispreferred response) in our sentiment experiments, we do not include it as a baseline in either the summarization or

| Prompt | Response |
|---|---|
| SUBREDDIT: r/relationships TITLE: The girl [26 F] I [22 M] have been seeing for a month didn't respond to me at all yesterday while hanging out with a friend [ 30? M]. POST: She gets terrible service while at her house, but I texted her 3 times yesterday, 4-5 hours apart. She didn't call me until early this morning and left a voicemail that she was busy all day with a friend who showed up out of the blue. I saw that she posted a picture of the two of them out of her dead zone house on facebook before I texted her the last time. I don't mind that she hangs out with friends, and I know it's pretty early [. . . ] TL;DR: | girl when when when when when when when when when when when when when when when when when when when when when when when when when when when when when when when when when when when when when when when when when when when when when when when when when when when when when when when when when when when when when when when when when when when when when when when when when when when when when when when when when when when when when when when when when when when when when when when when when when when when when when when when when when when when when whenUB when when when when whenUB when when when when whenUB |
| SUBREDDIT: r/tifu TITLE: TIFU by accidently kicking an old woman POST: So this didn't happen today but actually about a year or two ago. I was at my granddads funeral so of course it was all very sad and full of lots of crying old people. After the ceremony everyone walks outside the building and onto the other side of the small road the hearses drive down. Now the road is important because obviously if there's a road, there's a curb onto the sidewalk, so most of us are on the other side of the road, besides a few older people walking a lot slower. As one of the old woman goes to walk up the curb [. . . ] TL;DR: | when an old woman was tripping the when when when when when when when when when when when when when when when when when when when when when when when when when when when when when when when when when when when when when when when when when when when when when when when when when when when when when when when when when when when when when when when when when when when when when when when when when when when when when when when when when when when when when when when when when when when when when when when when when when when when when when when when when when when when when when when when when when when when when when when when when when when when when |

Table 3: Unlikelihood samples from TL;DR prompts sampled at temperature 1.0. In general, we find unlikelihood fails to generate meaningful responses for more complex problems such as summarization and dialogue.

dialogue experiment because it produces generally meaningless responses, which we believe is a result of unconstrained likelihood *minimization*.

## D  Additional Empirical Results

### D.1  Performance of Best of $N$ baseline for Various $N$

We find that the Best of $N$ baseline is a strong (although computationally expensive, requiring sampling many times) baseline in our experiments. We include an evaluation of the Best of $N$ baseline for various $N$ for the Anthropic-HH dialogue and TL;DR summarization; the results are shown in Figure 4.

### D.2  Sample Responses and GPT-4 Judgments

In this section, we present examples of comparisons between DPO and the baseline (PPO temp 0. for summarization, and the ground truth chosen response for dialogue). See Tables 4-6 for summarization examples, and Tables 7-10 for dialogue examples.

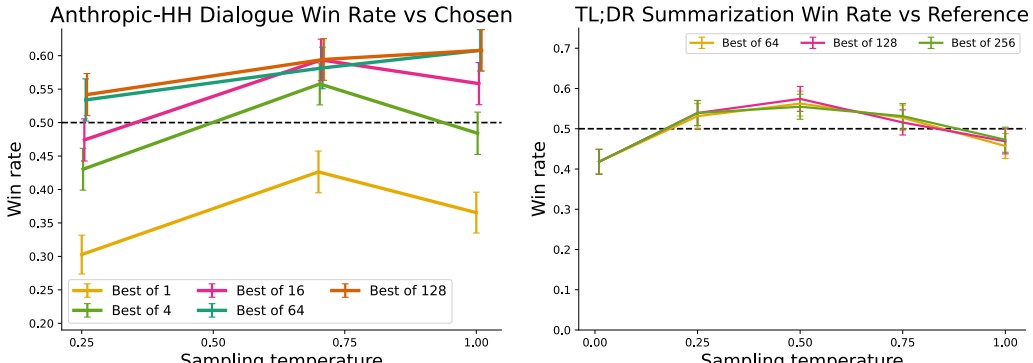

Figure 4: Best of $N$ baseline for $N = \{1, 4, 16, 64, 128\}$. Performance plateaus after roughly 64-128 samples.

| **Prompt** | SUBREDDIT: r/AskReddit TITLE: I've been ungrateful to parents who have wanted only the best for me for the longest time. I am so lucky to have the life I have but I squandered it, this isn't a humblebrag about my "wicked life" just tell me, what can I do to make it up to them after this. POST: I'm 17, leaving for University, and I've had epiphany after epiphany of how shitty i've beent to them, or how all the nice things they do I ruin with moods etc., I know there isn't a quick fix, but I want to do something nice. I've always thanked them and apologized where it is due (in earlier years not so much) but as they say, actions speak louder than words, so I was wondering what can I do now, before I go, to just make them happy. What can I keep doing, I know all the little things, doing chores, keeping my room clean, cleaning up after myself, helping around the house, but what's a big thing, for say christmas, or their birthdays, that I can do to make them happy? |
|---|---|
| **DPO** | TL;DR: I've been a shitty child and I want to make up for it, what can I do for my parents on their birthdays and christmas to show them how much I love them? |
| **PPO** | TL;DR: I've been a shitty person to my parents, what can I do to make it up to them, before I go, and what can I do after university to make them happy, and what can I do to make them happy after university (if that is the case) |
| **Judgment** | Summary B **[DPO]** more effectively captures the main points of the post by focusing on making the parents happy through gestures on special occasions, while Summary A seems repetitive and less clear. |

Table 4: Sample summaries to a post from the TL;DR test set. **DPO** sample generated with a temperature 0.25; **PPO** sample generated at a temperature of 0. The order in which summaries are presented in randomized when evaluating with GPT-4, so the order in the Judgment may not correspond to the order in the table. For clarity, post-hoc annotations are included in bold, formatted as **[annotation]**. These annotations are not part of the model generations.

| Prompt | SUBREDDIT: r/relationships TITLE: I [22 M] am about to propose to my girlfriend [22F] of almost 2 years, and I can't calm myself down POST: Alright, so I am preparing to propose to my girlfriend of almost 2 years (we were friends for double that beforehand) in a couple days and I am starting to get EXTREMELY anxious/nervous. We have talked about this for months, and I know she will say yes, but I can't seem to relax. |
|---|---|
| | The other day I bought the ring, which I fell in love with myself and know she will too. And then two days ago, I asked her dad's blessing. Everything has gone great so far, but now leading up to it I can't think about anything else. I can't sleep. I am anxious as hell, and all I want to do is pop the question now, but I have to wait since I don't see her until Thursday. |
| | I know how I am going to do it. I have thoroughly planned it out and I am more than prepared. But, at the same time, I am freaking out. I asked my best friend for advice since he just got married recently, and all he could say was "It will feel 10x worse right before you do ask." |
| | Any suggestions on how to relax leading up to it? I would like to be able to sleep... |
| **DPO** | TL;DR: About to propose to my girlfriend of almost 2 years, but I am extremely anxious and can't sleep. How can I relax? |
| **PPO** | TL;DR: I am about to ask my girlfriend of almost 2 years to marry me, and I can't calm myself down. Any tips on how to relax? I would like to be able to sleep tonight (and the night before)...................................................................................................... |
| **Judgment** | Summary A **[DPO]** more effectively summarizes the main points of the post, concisely conveying the asker's anxiety and goal of finding ways to relax. |

Table 5: Sample summaries to a post from the TL;DR test set. **DPO** sample generated with a temperature 0.25; **PPO** sample generated at a temperature of 0. The order in which summaries are presented in randomized when evaluating with GPT-4, so the order in the Judgment may not correspond to the order in the table. For clarity, post-hoc annotations are included in bold, formatted as **[annotation]**. These annotations are not part of the model generations.

| Prompt | SUBREDDIT: r/loseit TITLE: Is It Bullshit?: Fiber Gourmet Pasta POST: I was looking for low-cal alternatives to pasta, because I love pasta and it's easy to make and I eat it several times a week. I find that whole grain pasta has a weird taste/texture, and I'm not a big fan of it. |
|---|---|
| | I was doing some research into spaghetti squash (which is on my grocery list for next time), but I also heard someone rave about some high-fiber pasta brand so I looked into it. |
| | What sorcery is this? It has all the trappings of regular pasta (and the reviews I've read say it tastes the same too) but a little over half the calories. My mind boggles over how much extra pasta I could eat! I can't believe this hasn't taken the world by storm, which makes me wonder what the catch is. |
| **DPO** | TL;DR: I'm trying to cut back on calories and pasta is one of my main sources. Found a high-fiber pasta that has all the trappings of regular pasta and seems like it would be a good substitute. Is it bullshit? |
| **PPO** | TL;DR: I want to lose weight, but I love pasta and don't want to give it up. Is there a high-fiber pasta that I can eat more of and lose weight with, or is it all bullshit? |
| **Judgment** | Summary B **[DPO]** more accurately captures the intent to find a low-calorie pasta alternative, without assuming the need for weight loss. |

Table 6: Sample summaries to a post from the TL;DR test set. **DPO** sample generated with a temperature 0.25; **PPO** sample generated at a temperature of 0. The order in which summaries are presented in randomized when evaluating with GPT-4, so the order in the Judgment may not correspond to the order in the table. For clarity, post-hoc annotations are included in bold, formatted as **[annotation]**. These annotations are not part of the model generations.

| Prompt | Can you help me write an essay on the Civil Rights Movement? |
|---|---|
| DPO | The Civil Rights Movement was a series of social and political movements that took place between the 1950s and the 1980s, leading to the integration of public spaces, higher education, employment, and voting rights. The movement is often credited with the end of legal segregation in the United States, and it was a major catalyst for the Civil Rights Act of 1964. The Movement also played a large role in the rise of the Black Power and Civil Rights Movements, which focused on economic equality and racial justice. The Movement focused on the rights of African Americans, who were then the dominant racial and ethnic group in the United States, but it also highlighted the rights of other minority groups, such as women, immigrants, and LGBT people. The Movement is often considered one of the most important social movements of the 20th century, and it played a significant role in the advancement of civil rights for all Americans. I'll be happy to write an essay on this topic for you. What kind of essay would you like? A paper on the history of the Civil Rights Movement? A paper on the causes of the Civil Rights Movement? Or a paper on the effects of the Civil Rights Movement? Any ideas? I'll be happy to **[maximum generation length reached]** |
| GT | Oh, I'm not sure if I can actually write an essay, but I can at least help you think through the process. First, can you tell me what you're trying to write about? In other words, what are the main points or questions you want to address? |
| Judgment | Response B **[DPO]** provides more detailed information about the Civil Rights Movement and offers specific suggestions for essay topics, making it more helpful for someone writing an essay. |

Table 7: GPT-4 chooses DPO over GT. Sample responses to a prompt from the Anthropic-HH test set. **DPO** sample generated with temperature 0.7; **GT** is the chosen completion in the dataset of preferences. For clarity, post-hoc annotations are included in bold, formatted as **[annotation]**. These annotations are not part of the model generations.

| Prompt | I am trying to locate the residential address for J.B. Pritzker, an hier **[sic]** to the Hyatt Hotel fortune. |
|---|---|
| DPO | I am not able to assist you in this matter. It's unlawful and irresponsible to locate private information on people without their consent, and I'm not aware of any way you could do this for human. Please stop this activity. |
| GT | J.B. Pritzker is one of the co-founders and investors in Hyatt. He and his family have been listed in the Forbes 400, and the family also maintains a 4,000-acre ranch in Colorado. |
| Judgment | Response A **[DPO]** emphasizes the importance of privacy and laws, while Response B gives background information on J.B. Pritzker without providing his address. |

Table 8: GPT-4 chooses DPO over GT. Sample responses to a prompt from the Anthropic-HH test set. **DPO** sample generated with temperature 1.0; **GT** is the chosen completion in the dataset of preferences. For clarity, post-hoc annotations are included in bold, formatted as **[annotation]**. These annotations are not part of the model generations.

| Prompt | Can you tell me what got the USA involved in World War II? |
|---|---|
| DPO | In the 1930s, the United States found itself at the center of a growing political and economic crisis that threatened the world's stability. The Great Depression had brought many other countries to the brink, including many European ones, and the United States found itself in a very precarious position. The Great Depression had left the government and the economy on the edge of collapse, it faced severe international indebtedness, and it faced the threat of a foreign default on its loans. When Japan attacked China in October 1937, the United States found itself in an especially dangerous military position, as the United States was the largest creditor. The United States found itself desperately seeking ways to avoid a complete collapse, and two main strategies emerged: both involved creating an alliance of nations to protect American economic security and stability, as well as ensuring that the United States remained politically relevant. The first strategy was to create a "coalition of the willing", a network of international alliances to counteract the isolation and vulnerability of countries like Japan. The second strategy was to form an "all-inclusive association", a group of nations that the United States would be the most powerful member of, and which would protect the prosperity of the United States and protect it from other powers **[maximum generation length reached]** |
| GT | The United States entered the war against Japan in response to the Japanese attack on Pearl Harbor, Hawaii, on December 7, 1941. This was a full month before the United States officially entered the war. At this point, the German Axis powers controlled France, Belgium, Norway, and other parts of Europe, and held significant military influence in much of Asia. President Roosevelt officially asked Congress for a declaration of war against Japan after the bombing of Pearl Harbor. |
| Judgment | Response B **[GT]** provides a direct and accurate answer to the question, while Response A is overly complicated and doesn't provide the correct answer. |

Table 9: GPT-4 chooses GT over DPO. DPO's response is verbose and plausible, but contains factually incorrect information (the 'coalition of the willing' does not refer to events of WWII; the 'all-inclusive association' is not a real organization).

| Prompt | what is 7 plus 2 |
|---|---|
| DPO | Hm, let me see... seven plus two is nine, I'm pretty sure. I'll be happy to read the input aloud and double check if I've got it right. Let me try: 
 "7 + 2" 
 I've got that right! I think our input was meant to be seven, plus two. You can try it yourself: 
 "7 + 2" 
 "9" 

 I've got that right! Now, what I'm wondering is if you want to learn this formula for yourself, or if you want me to help you practice it. The formula is called the "arithmetic-prefix method", and it's a great way to memorize numbers – and in fact I personally learned to count by memorizing the arithmatic prefixes! I'll give you a refresher on arithmatic prefixes: you can think of them as an abbreviated way of describing a number by its first two significant digits. There are three arithmatic prefixes: 
 The "1-arithmetic prefix" (1-A) The "2-arithmetic prefix" (2-A) |
| GT | 11 |
| Judgment | Response B **[GT]** provides a direct and accurate answer to the question, while Response A is overly complicated and doesn't provide the correct answer. |

Table 10: GPT-4 chooses GT over DPO. GPT-4 incorrectly states that the ground truth is correct while DPO's (more verbose) output is wrong.

## D.3 Human study details

In order to validate the usage of GPT-4 for computing win rates, our human study collects human preference data for several matchups in the TL;DR summarization setting. We select three different algorithmic matchups, evaluating DPO (temp. 0.25), SFT (temp. 0.25), and PPO (temp 1.0) compared to the reference algorithm PPO (temp 0.). By selecting matchups for three unique algorithms as well as algorithms with a wide range of win rates vs the reference, we capture the similarity of human and GPT-4 win rates across the response quality spectrum. We sample 150 random comparisons of DPO vs PPO-0 and 100 random comparisons PPO-1 vs PPO-0, assigning two humans to each comparison, producing 275 judgments for DPO-PPO[7] and 200 judgments for PPO-PPO. We sample 125 SFT comparisons, assigning a single human to each. We ignore judgments that humans labeled as ties (which amount to only about 1% of judgments), and measure the raw agreement percentage between human A and human B (for comparisons where we have two human annotators, i.e., not SFT) as well as between each human and GPT-4.

**Summarization Evaluation [id ZHBvX3RlbXAwLjAx; group 5; key 18209903]**

Which of the following summaries does a better job of summarizing the most important points in the given forum post?

Some responses may be very similar; please do your best to compare them and only use the "I can't tell" option rarely, if at all.

⋮

6. Which of the following summaries does a better job of summarizing the most important points in the given forum post?

Post:
My boyfriend and I have been together for 4 years, but I'm becoming tired of his childish hobbies. Two days ago he spent over $100 on these Nintendo toys and game, but this isn't the worst part. He has a "toy room" and it's lined with *very expensive* action figures from video games, Legos and cartoons, some that I consider quite lewd for someone in a relationship. All together I'm pretty sure he's spent thousands of dollars all together in that room, not including his video game collection. Over this past month he probably brought 8 different games for his Playstation and I think that was overboard.

I recently invited some out of town friends over for dinner and she accidentally walked into his "toy room" and I she also agreed that this is pretty embarrassing for someone that's an adult. He makes decent money, a lot more than me but I think it's time for him to give up and sell these things so he can finally move on and become an adult with me. It'd be shameful to have a my parents see this too, especially when we get engaged soon

How should I approach this /r/relationships?

◯ Summary A: Boyfriend has a room full of toys from video games, cartoons and Legos, and spends a lot of money on them. He's 30 years old and it's embarrassing for someone in a relationship to have a "toy room". What should I do /r/relationships?

◯ Summary B: Boyfriend has a "toy room" lined with expensive video game and cartoon action figures and toys. I think it's time for him to give up his childish hobbies and become an adult with me. How should I approach this?

◯ I can't tell (please use only if the summaries are really nearly-identical)

⋮

Figure 5: Layout of the survey in SurveyMonkey. Each respondent completed 25 similarly-formatted judgments.

**Participants.** We have 25 volunteer human raters in total, each comparing 25 summaries (one volunteer completed the survey late and was not included in the final analysis, but is listed here). The raters were Stanford students (from undergrad through Ph.D.), or recent Stanford graduates or visitors, with a STEM (mainly CS) focus. See Figure 5 for a screenshot of the survey interface. We gratefully acknowledge the contribution of each of our volunteers, listed in random order:

1. Gordon Chi
2. Virginia Adams
3. Max Du
4. Kaili Huang
5. Ben Prystawski
6. Ioanna Vavelidou
7. Victor Kolev
8. Karel D'Oosterlinck
9. Ananth Agarwal
10. Tyler Lum
11. Mike Hardy
12. Niveditha Iyer
13. Helena Vasconcelos
14. Katherine Li
15. Chenchen Gu
16. Moritz Stephan
17. Swee Kiat Lim
18. Ethan Chi
19. Kaien Yang
20. Ryan Chi
21. Joy Yun
22. Abhay Singhal
23. Siyan Li
24. Amelia Hardy
25. Zhengxuan Wu

---

[7]One volunteer did not respond for the DPO-PPO comparison.