# OpenReview forum: "Direct Preference Optimization: Your Language Model is Secretly a Reward Model"
_NeurIPS.cc/2023/Conference — NeurIPS 2023 oral_

### Official Review · Reviewer_eRhN · 2023-07-04

**Soundness:** 4 excellent
**Presentation:** 4 excellent
**Contribution:** 4 excellent
**Rating:** 8
**Confidence:** 3

**Summary:**

This paper tackles a very important problem in LLM training, i.e., how to simplify the complex process of reinforcement learning from human feedback? RLHF is different to use, it is a complex and often unstable procedure, first fitting a reward model that reflects the human preferences, and then fine-tuning the LLM using RL to maximize this estimated reward. The authors propose DPO, a training paradigm for training language models from preferences without reinforcement learning. DPO identifies a mapping between language model policies and reward functions that enables training a language model to satisfy human preferences directly, with a cross-entropy loss, without reinforcement learning. The experimental results show that DPO can fine-tune LLMs as well or better than existing algorithms.

**Strengths:**

This article addresses a crucial issue in LLM training, i.e., how to perform alignment in a simpler way without using RL. The proposed DPO is remarkably simple yet effective, and it exhibits good theoretical properties. I believe DPO is highly significant for researchers in the field and will serve as a powerful tool for their work.

**Weaknesses:**

This is a very solid piece of work. The proposed method is simple yet effective. I don't have any particular concerns or issues with it.

**Questions:**

None

**Limitations:**

Yes

---

> ### Author Rebuttal · Authors · 2023-08-10
>
> Thank you for your detailed summary of the paper, your feedback, and believing that our work is highly significant for researchers!

---

> > ### Comment · Reviewer_eRhN · 2023-08-11
> >
> > I have read the author's response, and I maintain my score and recommend accepting this paper.

---

### Official Review · Reviewer_cpnK · 2023-07-06

**Soundness:** 3 good
**Presentation:** 3 good
**Contribution:** 4 excellent
**Rating:** 7
**Confidence:** 5

**Summary:**

This paper presents a new policy optimization algorithm with human preferences, called Direct Preference Optimization (DPO), which eliminates the need for explicitly fitting a reward model. The authors formulate the process of RLHF into a single stage policy learning by leveraging a mapping between reward functions and optimal policies. The main contribution of this paper is that they theoretically derive a new objective function that is equivalent to existing RLHF methods but can be simply learned by a single objective, and experimentally shows that it has better performance.

**Strengths:**

- This paper presents a new algorithm that can simply learn the policy from a given human preference dataset without explicit reward learning.
- This paper not only experimentally shows better results than RLHF, but also provides theoretical connections with RLHF.

**Weaknesses:**

- Since the reward model is not explicitly learned and the policy is learned only with the given human preference dataset, there is a possibility that the generalization performance for unseen prompts is relatively low compared to the existing RLHF.
- Experiments are insufficient to show that DPO is more stable than conventional RLHF. In order to show that the DPO can be learned more stably, it seems that the learning curve should be provided together with the results shown in the experiment.
- A detailed description of the experiment is lacking. (ex. amount of human preference dataset used in the experiment)
- Depending on the amount of human preference dataset, DPO may have better or worse performance than RLHF, but there is no analysis or explanation about this.

**Questions:**

- In Figure 2 of the experiment results, why does DPO perform better than PPO-GT even though PPO-GT uses a true reward function?
- The values of the target KL used in PPO-GT are large (target KL $\in$ {3, 6, 9, 12}). Is there a reason why the hyperparameter search range is set this way? In the previous paper (ex. [1]), hyperparameter search was done in a much wider range (target KL $\in$ {0.02, 0.05, 0.1, inf}), and there were good results at low values, so I wonder if the results of the PPO-GT used in the paper were suboptimally learned.
- In the results according to the sampling temperature, why does DPO have robust results to the sampling temperature?
- How does the fluency (or naturalness) of the generated sentences from the learned policy according to training iteration change in the DPO experiment? In the case of RLHF, the fluency of the generated sentence shows a result that is sensitive to the coefficient of the KL divergence term (i.e. Fluency deteriorates as learning progresses in RLHF algorithm). I wonder if DPO does not have a similar problem.
- Also, it was mentioned in the paper that DPO learning is stable regardless of hyperparameter beta, but I wonder if learning according to various beta values is stable without degeneration issues (i.e. degrading the fluency).
- Unlikelihood learning (i.e. decreasing the likelihood) objectives are easy to degrade fluency (or naturalness), but how is it possible to stably learn while including unlikelihood learning objectives in DPO? It seems that the main difference between the equation for the gradient of the DPO and unlikelihood learning is that the weight term is multiplied. What exactly does the weight do to make learning stable?
- What does the star mean in the right plot of Figure 2?

[1] Rajkumar Ramamurthy and Prithviraj Ammanabrolu et al, Is Reinforcement Learning (Not) for Natural Language Processing: Benchmarks, Baselines, and Building Blocks for Natural Language Policy Optimization, ICLR 2023

**Limitations:**

- Since the reward model is not explicitly learned and used, generalization to unseen prompts may not work well.
- A comparison of RLHF and DPO in various experimental settings is lacking.
- As mentioned in Questions and Weaknesses, it seems that there are still some parts that are not clearly explained or verified.
- Details of the experimental setup are omitted.

---

> ### Author Rebuttal · Authors · 2023-08-10
>
> Thanks for your helpful feedback and questions!
>
> ### Generalization of DPO vs existing RLHF
>
> First, we note that all of the evaluations in the paper already compute test win rates on unseen test prompts.
>
> We performed a follow-up experiment to assess the performance of PPO and DPO under distribution shifts. We evaluate the summarization policies trained on Reddit TL;DR data on the CNN/DailyMail dataset, and evaluate the win rate of each policy's sample vs the ground truth summary. We use the sampling temperatures that performed best for TL;DR. For space reasons, please see the general comment for full experimental details.
>
> **Overall, DPO shows superior performance to PPO for OOD inputs as well:**
>
> **DPO-0 winrate vs ground truth: 0.359 (+/- 0.030)**
>
> **DPO-0.25 winrate vs ground truth: 0.309 (+/- 0.029)**
>
> **PPO-0 winrate vs ground truth: 0.258 (+/- 0.027)**
>
> **PPO-0.25 winrate vs ground truth: 0.230 (0.026)**
>
> ### DPO stability
>
> Thanks for the suggestion! To more clearly illustrate DPO's stability across training runs, we trained 4 DPO models using EleutherAI/Pythia-1b on the Anthropic-HH dataset. Please find the training loss curves in Fig. 2 in the rebuttal figure pdf and the classification accuracy of the implicit DPO reward function in Fig. 4. We find that these learning curves are very consistent across training runs. Further, we evaluate the final GPT-4 win rate of each policy against the chosen response for unseen prompts in the test set. The win rates for the final policy from each training run were 0.426, 0.410, 0.414, 0.438 (standard dev: 0.0110). In comparison, sampling from the first policy four times, changing only the random seed used for sampling, we observe a similar distribution of scores, with slightly higher standard deviation: 0.426, 0.412, 0.406, 0.441 (standard dev: 0.0135). These results suggest that the variance in performance of DPO policies across different training seeds is less significant than the variance due to simply using different random seeds for sampling at test time.
>
> Additionally, we also experiment with how the \beta hyperparameter controls the KL-divergence, we find that the \beta hyperparameter very reliably controls the KL-divergence of the final policy, as shown in Fig. 1 in the rebuttal figure pdf. In contrast, PPO implementations typically require an adaptive beta (based on a target KL divergence) because the same fixed beta can lead to very different KLs for different training runs.
>
> ### Size of human preference dataset
> Thanks for bringing up the missing details. We use publicly available preference datasets for summarization (HuggingFace dataset CarperAI/openai_summarize_comparisons) with 92.5k comparisons and Anthropic HH dialogue (HuggingFace dataset Anthropic/hh-rlhf) with 161k comparisons. We will include these and other details about the preference datasets in the camera ready version. In our experiments, DPO and PPO use the same preference datasets (and thus, the same amount of preference data). While our new experiment evaluating the OOD performance of PPO and DPO gives some reason for optimism about DPO's data efficiency, directly comparing the performance of DPO and PPO as the amount of preference data is varied is an important question for analysis, which we defer to future work. Thank you for this suggestion!
>
> ### Why does DPO perform better than PPO-GT even though PPO-GT uses a true reward function?
> In the controlled sentiment generation problem, we find that the learned PPO reward can achieve accuracy well over 90%. Hence we believe the discrepancy between DPO and PPO is not due to issues with reward modeling, but optimization of the reward. PPO only approximately optimizes the KL-constrained reward problem, while DPO samples from the closed-form optimal policy (without approximation). We hypothesize that PPO is less efficient than DPO, even with the ground truth reward, due to this noisy optimization.
>
> ### KL target values scaling
> Figure 2 reports sequence-level KL, which adjusting for sequence length results in per-token KLs of about 0.05, 0.1, 0.15 and 0.2, comparable to those used in Ramamurthy et. al.
>
> ### Fluency during training
> DPO has a very stable relationship between the KL divergence of the final policy and the hyperparameter $\beta$, as shown in the attached Fig. 1. We do not observe any degeneration as training progresses, however, in the limit of $\beta$ going to zero, the KL constraint vanishes, and DPO would purely maximize the (implicit) learned reward. In this case, optimizing a reward function trained on a small preference dataset would likely show some deterioration.
>
> ### Unlikelihood stability
> Indeed, unlikelihood objectives can degrade fluency. As we note in Section 4, the weight scaling the unlikelihood updates in DPO in would be < 1 for all comparisons, but for comparisons where the preferred completion has a higher reward than the dispreferred completion, the weight can be substantially lower, effectively stopping learning on those examples. Thus, the adaptive weights only change the model whenever the preference pair is ordered incorrectly under the LM, which we hypothesize leads to the stable training regime. Further, as beta is larger, this weight goes to zero more rapidly as the example is learned, effectively stopping optimization more quickly. Nonetheless, we cannot rule out the possibility that carefully tuned unlikelihood methods can also be successfully used for fine-tuning LM on comparison pairs.
>
> ### Figure 2 star
> Stars represent the win rates as computed in human evaluation, and we include 3 stars corresponding to the win rates for DPO @ 0.25, SFT @ 0.25 and PPO @ 0.0, all compared against PPO @ 0.0. We will clarify this in the figure caption.

---

> > ### Comment · Area_Chair_rtVu · 2023-08-18
> > **Thanks for authors' rebuttal!**
> >
> > Reviewer cpnK, did the authors address your concerns about generalization performance for unseen prompts, DPO stability, as well as other concerns? Thanks.

---

> > ### Comment · Reviewer_cpnK · 2023-08-20
> > **Response to the rebuttal**
> >
> > Thank you for the rebuttal and the additional experiments. Since the authors have addressed most of my main concerns (generalization performance for unseen prompts and DPO stability), I would like to raise my score. I thank the authors for their response.

---

### Official Review · Reviewer_GfFq · 2023-07-06

**Soundness:** 3 good
**Presentation:** 2 fair
**Contribution:** 4 excellent
**Rating:** 8
**Confidence:** 3

**Summary:**

The authors propose DPO, a method to fine-tune a language model to human preferences that does not use reinforcement learning or learn a reward model, but rather, directly optimizes the language model with preference data.

Through a change of variables, the authors express the reward function in terms of the optimal policy $\pi*$ and $\pi^{ref}$. The authors write the human preference distribution in terms of $\pi*$ and $\pi^{ref}$, and thus a maximum likelihood objective can be written in terms of $\pi_\theta$ and $\pi^{ref}$. This formulation allows for a simple DPO gradient update.

The method is equivalent to fitting a reparameterized Bradley-Terry model. With mild assumptions, DPO does not constrain the class of learned reward models.

Experimental results are run on three different open-ended generation tasks. Baselines include PPO, zero-shot and few-shot prompting, a SFT model, and Preferred-Ft. DPO’s reward/KL tradeoff dominates that of PPO. Furthermore, win-rates of DPO compared to other models (or test-set labels) beat baselines.



**Strengths:**

The paper is a significant contribution in a high impact area -- namely, fine-tuning language models to human preferences. DPO opens a new paradigm of learning human preferences without RL, and is a viable and performant alternative to the RLHF pipeline, reducing computational demands, and requires little tuning of hyperparameters

DPO is grounded theoretically, equivalent to fitting a reparameterized Bradley-Terry model. The DPO update is simple and interpretable. Theorem 1 shows we do not necessarily constrain ourselves in terms of the reward model after reparameterization.

Experimental results are good. DPO learns a better Reward / KL tradeoff compared to baselines in the sentiment generation task. DPO also has excellent win-rates compared to a number of baselines as evaluated with GPT-4 on the summarization and single-turn dialogue tasks. Human evaluators show as high correlations with each other as compared with the GPT-4 evaluation.

**Weaknesses:**

Single-turn dialogue experimental results did not compare DPO with a RLHF baseline tuned from the same base model, and it appears that helpful-rejection-sampled data from the Anthropic HH dataset was used, a relatively weaker baseline compared to the helpful-online dataset.

There are very few details on the experimental setup for the IMDB Sentiment Generation task.

Improvements to the clarity of the paper can be made, including more detailed descriptions of experimental setup in the Appendix.

**Questions:**

Please provide experimental details for the IMDB Sentiment Generation task.

Please clarify what dataset, (specifically, whether the “helpful-online” or “helpful-rejection-sampled” data) was used for the single-turn dialogue experiment?

Why does win-rate in the single-turn dialogue task increase with higher sampling temperature in Figure 3?

**Limitations:**

Assumes we use Plackett-Luce models to model preferences.

---

> ### Author Rebuttal · Authors · 2023-08-10
>
> Thanks for assessing our paper to be a significant contribution, and for your feedback!
>
> ### Single-turn dialogue experimental results did not compare DPO with a RLHF baseline…
>
> Please, see Fig. 3 in the rebuttal pdf for an additional baseline to our dialogue experiments, the oracle best-of-N method that uses a reward model learned on all the preference data to pick the best of N samples from the SFT model. We observe that DPO performs comparably to the best-of-N method. Note that best-of-N is computationally infeasible as it requires generating N samples at test-time (for example, 128) and choosing the best one, but this baseline is often used as an oracle proxy for PPO [1] and achieves comparable performance (Table 2 in [2]). We use best-of-N because it is easy to implement, as strong as PPO, and public RLHF checkpoints that we found for Anthropic use undocumented training splits and/or skip the SFT phase, which is non-standard and makes fair comparison with DPO difficult.
>
> [1] Scaling Laws for Reward Model Overoptimization. Gao, Schulman, Hilton.
>
> [2] AlpacaFarm: A Simulation Framework for Methods that Learn from Human Feedback, Dubois, et. al.
>
>
> ### There are very few details on the experimental setup for the IMDB Sentiment Generation task…
>
> Thanks for the suggestion! Here are additional details about the IMDB task, which we will add to Section 6.1 of the next revision of the paper:
>
> Following the trl library, the prompts are prefixes from the IMDB dataset of length 2-8 tokens. We use the pre-trained sentiment classifier "siebert/sentiment-roberta-large-english" as a ground-truth reward model and GPT2-large as a base model. We use these larger models as we found the default ones to generate low-quality text and rewards to be somewhat inaccurate.  We first use supervised fine-tuning on a subset of the IMDB data for 1 epoch. We then use this model to sample 4 completions for 25000 prefixes and create 6 preference pairs for each prefix using the ground-truth reward model. The RLHF reward model is initialized from the GPT2-large model and trained for 3 epochs on the preference datasets, and we take the checkpoint with the highest validation set accuracy. The “TRL” run uses the same hyper-parameters as the trl library implementation. Our implementation uses larger batch samples of 1024 per PPO step.
>
> ### Please clarify what dataset, (specifically, whether the “helpful-online” or “helpful-rejection-sampled” data) was used for the single-turn dialogue experiment?
>
> We used the entire Anthropic/hh-rlhf available on HuggingFace. We will add the details to the paper.
>
> ### Why does win-rate in the single-turn dialogue task increase with higher sampling temperature in Figure 3?
> For low-temperature (0.25) evaluation in the dialogue task, we observe a significant number of responses that get stuck in repetition loops, a common failure mode of smaller LMs being sampled at low temperature. For the two higher-temperature (0.7, 1.0) evaluations, the win rates are within one standard error of each other. We will note this observation of repetition loops at low temperature in Section 6.2 of the revised paper.

---

> > ### Comment · Reviewer_GfFq · 2023-08-15
> >
> > I have read the response as well as the common points from the authors. I increase my score to 8.

---

### Official Review · Reviewer_FksS · 2023-07-08

**Soundness:** 4 excellent
**Presentation:** 3 good
**Contribution:** 4 excellent
**Rating:** 8
**Confidence:** 4

**Summary:**

RLHF is generally done by training a reward model on a dataset of preferences and then using on-policy RL to finetune a language model with the reward model. This work proposes to bypass explicit learning of a reward model and finetune the language model directly on the preference dataset. This is conceptually similar to offline RL but the authors derive a direct preference objective such that the training is supervised not RL-based.

Using a standard preference model, a supervised objective (DPO) is derived, theoretically justified, and compared to PPO. The authors test their method on three tasks: sentiment completion (IMDB), summarization (as in Stiennon et al), and single-turn dialogue with Anthropic's HH dataset. They find that DPO achieves higher reward while staying closer to the reference model (KL) in the IMDB task. Using GPT-4 as a proxy for human judgement on summarization, DPO can match PPO's performance while being less sensitive to sampling temperature. They validate GPT-4 with human annotators and find that DPO is even slightly more preferred over PPO. On dialogue, they find that DPO outperforms supervised and re-ranking (again using GPT-4 eval) but do not compare against PPO.

**Strengths:**

The overall idea is relatively simple but the justification and theorems provide an excellent basis. The paper is timely and gives an excellent alternative to RLHF, that should be notably more efficient. The paper is clearly written and has the potential to be very impactful if used at scale in modern methods. GPT-4 evaluations and the follow-up human evaluations are strong and demonstrate both improved performance and robustness which can be notably bad in RLHF. The authors include the minimal code for DPO in the appendix which is commendable and many GPT-4 evaluation details for excellent reproducibility. The presentation is also very clear and breaking down the DPO loss into its components with clear language is an excellent addition.

**Weaknesses:**

The main weakness is the unexplored main difference between DPO and PPO: generalization. The authors note it in limitations but since PPO is trained on-policy and DPO is trained offline, looking at generalization outside the dataset would be helpful. The work, as is, feels sufficient without those experiments but the difference does play into an inconsistency with some theorems. The other major issue are some incongruenties in the evaluation that the authors should respond to.

The proofs and theorems contain a hidden assumption that is not explicitly stated. In the formulations for PPO, the optimization uses $y \sim \pi_\theta$ whereas in the formulations for DPO $y \sim D$ i.e. DPO's algorithm is offline and the model learned from its implicit reward has guarantees over the dataset. In contrast, PPO's algorithm is learned over a new set of datapoints sampled from the model during training. This means that Equation 3 is maximized over a different set of data points than Equation 7. So even though Theoerem 1 is correct, Lemma 2 does not fully encompass this situation. With offline DPO vs on-policy PPO, equivalent reward models could induce different optimal policies since they are being optimized over a different set of points. The section seems to imply that PPO and DPO can induce the same optimal policies because of the ambiguity with $y$. Clarifying this would be helpful. This also means that section 5.2 and the substitution used to make Equation 10 is not exactly accurate as the optimal policy for PPO is not the same as the optimal policy for DPO. I still feel the point of section 5.2 is reasonable but the authors should explicitly note the issue with the substitution.

There are issues with each of the different evaluations, even though the overall picture does suggest that the method is sound. I will increase my score if the authors simply clarify some questions about the evaluations (not even necessary to do changes / re-run experiments)

For controlled sentiment (IMDB), the authors do not give many details, seem to deviate from previous work on the task without explanation. The GRUE benchmark (RL4LMs) specifically uses GPT-2 base and finds that models should not be supervised-finetuned before RL training. In contrast, the authors use GPT-2 large (not even GPT-2 medium) and do supervised finetuning before PPO. No details of the baseline are given except for the graph legend which implies the PPO baseline comes from the `trl` library, although the authors only claim to use the `trlx` library. It is also unclear whether the evaluation is over the train set or the test set. Since this task is the only one where the authors show the tradeoff between reward achieved and KL, these details seem important.

For summarization, the issue is relatively minor. GPT-4 may be an unfair evaluator (see Large Language Models are not Fair Evaluators) and the work would benefit from evaluating samples in both positions (i.e. option A then B and option B then A)

For Anthropic's HH, the authors claim there is no SFT model available but there are many available pretrained models for this benchmark that the authors do not compare against. The most prominent example being a GPT-J 6B model `reciprocate/ppo_hh_gpt-j` hosted on huggingface which also has wandb runs and repro code available on the `trlx` library. The authors have chosen Pythia 2.8B when there exists a similar Pythia 1.6B PPO-trained on the HH dataset and it is unclear why the authors do not compare to this model or another PPO-trained model.


**Questions:**

What are the compute differences between DPO and PPO? It feels like DPO should be much more efficient and this information would only strengthen your work.

Why do you not compare against ILQL for sentiment? As an offline RL method, it seems the closest RL comparison to DPO

For reproducibility, what exact API are you using for the GPT-4 evaluation (e.g. GPT-0314)

Can you move Figure 2 and 3 to the next pages so it is closer to the text that references it?

**Limitations:**

The authors did quite well to address limitations in the paper, I believe they covered most of my concerns there.

---

> ### Author Rebuttal · Authors · 2023-08-10
>
> Thanks for appreciating our work and providing a great summary for it!
>
> ### DPO vs PPO generalization.
>
> We performed a follow-up experiment to assess the performance of PPO and DPO under distribution shifts. We evaluate the summarization policies trained on Reddit TL;DR data on the CNN/DailyMail dataset, and evaluate the win rate of each policy's sample vs the ground truth summary. We use the sampling temperatures that performed best for TL;DR. For space reasons, please see the general comment for full experimental details.
>
> **Overall, DPO shows superior performance to PPO for OOD inputs as well:**
>
> **DPO-0 winrate vs ground truth: 0.359 (+/- 0.030)**
>
> **DPO-0.25 winrate vs ground truth: 0.309 (+/- 0.029)**
>
> **PPO-0 winrate vs ground truth: 0.258 (+/- 0.027)**
>
> **PPO-0.25 winrate vs ground truth: 0.230 (0.026)**
>
> ### PPO $y \sim \pi_\theta$ vs DPO $y \sim D$
>
> We note that Eq. 3 and Eq. 7 represent different stages of RLHF. Eq. 3 shows the objective of the policy optimization stage, while Eq. 7 shows the reward learning stage (using the DPO parameterization). Thus the interpretation of y is different in these equations. The DPO reward parameterization allows us to extract the optimal solution to the problem in Eq. 3 exactly, in closed form; this parameterization simply means that we don't have to perform any additional RL optimization (such as online learning with PPO), which conventional RLHF does.
>
> ### DPO vs PPO optimal policy for finite data
>
> In the limit of infinite data and perfect optimization, the policy found by PPO will be equivalent to the DPO policy, which is the optimal policy (wrt the learned DPO reward) in Eq. 4, which holds for all prompts & answers (this assumes the PPO reward model agrees with the implicit DPO reward model; Theorem 1 and our empirical findings show that the two reward model classes have equivalent expressiveness, and our empirical findings suggest that the two parameterizations do achieve very similar classification accuracy on held out preference pairs). However, in the case of finite data or imperfect optimization, PPO might not recover the theoretically optimal policy that DPO does (Eq. 4). Our new generalization experiment in the summarization setting further suggests that the DPO policy generalizes at least as well, if not better than, the PPO policy. Finally, Lemma 2 only states equivalence of the optimal policy, but suboptimal policies found from finite data or imperfect optimization may indeed differ for two reward functions in the same equivalence class.
>
> ### Substitution in Equation 10
>
> Thanks for pointing this out; this is a typo. The optimal policy is the one induced by Eq. 4, not Eq. 7. We apologize for the confusion and will correct the mistake in the final version of the paper. The result (closed-form optimal policy) of Eq 4 holds on all prompts and answers, given a reward function. We agree that the paper would benefit from this discussion, and the points raised about the difference in the PPO / DPO problem settings and the optimal policies being different. Specifically, we will revise Section 5.2 to note the discrepancy between optimal policies.
>
> ### IMDb details
>
> Thanks for raising the concern about the missing details. We provide the details of the controlled sentiment generation task in the common response, along with the details for the PPO baseline. We follow the TRL library which first does an SFT step before running PPO. This is necessary since the experiment uses shorter prompts and the model might deviate from the movie review task. The evaluation is done on the test set. We will add all the details to the paper in the next revision.
>
> ### GPT-4 evaluation sample ordering
>
> Our present evaluation protocol randomly flips options A and B for every evaluation prompt for this reason. We make a brief note of this in Appendix C.1, but will certainly clarify in the main text.
>
> ### Anthropic HH baselines
>
> We have added a comparison of DPO to the oracle best-of-N method using a reward model learned on all the preference data; see Fig. 3 in the attached rebuttal figure pdf. We observe that DPO performs comparably to the best-of-N method. Note, this method is computationally infeasible as it requires generating N samples at test-time (for example, 128) and choosing the best one, but, this baseline is often used as an oracle proxy for PPO [1][2]. We use best-of-N because it is easy to implement, as strong as PPO, and public RLHF checkpoints that we found for Anthropic use undocumented training splits and/or skip the SFT phase (e.g., the reciprocate/ppo_hh_gpt-j checkpoint you mentioned) which is non-standard and makes fair comparison with DPO difficult.
>
> [1] Scaling Laws for Reward Model Overoptimization. Gao, Schulman, Hilton.
>
> [2] AlpacaFarm: A Simulation Framework for Methods that Learn from Human Feedback, Dubois, et. al.
>
> ### PPO vs DPO compute
> Depending on design choices DPO can require as little as half as much accelerator memory than PPO, since it does not maintain separate value function and reward models. In terms of run times, we found that DPO training requires about 5-8 times less compute time than the standard RLHF pipeline, depending on the model and dataset.
>
> ### ILQL baseline
>
> It is possible to use many different RL algorithms to align a model with the learned reward function and in this work we focused on PPO due to its prevalence. Concurrent work (Appendix D3 in [1]) evaluated several offline RL algorithms on the Anthropic HH datasets using a learned reward model and found that they yielded little to no improvement.
> [1] Fine-Tuning Language Models with Advantage-Induced Policy Alignment, Zhu et. al.
>
> ### GPT-4 API details
>
> We used the `gpt-4-0314` for evaluation. We will add the details to the experiments section of the paper.
>
> ### Figure 2/3 positioning
> Thanks for the suggestion! We will reformat the paper so that the figures are closer to the text referencing it.

---

> > ### Comment · Reviewer_FksS · 2023-08-13
> > **Re: rebuttal**
> >
> > I've read the rebuttal and the authors have addressed my concerns. I am increasing my score.
> >
> > **Generalization** these are very strong result and do suggest that DPO has some of the generalization properties. I would encourage you to add this to the main paper if possible.
> >
> > **Lemma 2** I agree that this is true in the limit of infinite data. I assumed that you are using the optimal policies given a finite dataset. If you can specify infinite data, then I have no issue with this.
> >
> > **Anthropic best of n** This is a sufficiently strong baseline and I think it covers my minor issues with evaluation here.
> >
> > **ILQL** I was not aware of this result, thank you for referencing it

---

### Author Rebuttal · Authors · 2023-08-10

We appreciate the detailed feedback provided by the reviewers. We address a few common points in this response. All other questions are addressed in reviewer specific responses.

## Re: Generalization of PPO and DPO

First, we note that all of the evaluations in the paper compute test win rates on unseen test prompts.

To further assess the performance of PPO and DPO under distribution shifts, we evaluated the PPO and DPO policies from our Reddit TL;DR summarization experiment on a different distribution, news articles in the test split of the CNN/DailyMail dataset, using the best sampling temperatures from TL;DR (0 and 0.25). We computed the GPT-4 win rate against the ground-truth summaries in the datasets, using the same GPT-4 (C) prompt we used for Reddit TL;DR, but replacing the words "forum post" with "news article". We found that for this new distribution, DPO continues to outperform the PPO policy by a significant margin:

**DPO-0 winrate vs ground truth: 0.359 (+/- 0.030)**

**DPO-0.25 winrate vs ground truth: 0.309 (+/- 0.029)**

**PPO-0 winrate vs ground truth: 0.258 (+/- 0.027)**

**PPO-0.25 winrate vs ground truth: 0.230 (0.026)**

While it is not a comprehensive evaluation of generalization of different RLHF algorithms, **this experiment suggests that DPO policies can generalize similarly well to PPO policies, even without training on the additional unlabeled Reddit TL;DR prompts that PPO uses**. We will include this experiment in our next revision, and defer a more extensive investigation to future work.

## Re: Baselines on Anthropic-HH dialogue

Some reviewers asked about additional baselines for the dialogue experiment.

We have added a comparison of DPO to the best-of-N method (i.e., generate N samples, return the one with highest reward under the reward model) using a reward model learned on all the preference data; see Fig. 3 in the attached rebuttal figure pdf. **We observe that DPO performs comparably to the strong best-of-N baseline.** Note, this method is computationally infeasible as it requires generating N samples at test-time (for example, 128) and choosing the best one, but, this baseline is often used as an oracle proxy for PPO [1][2]. We use best-of-N because it is easy to implement, as strong as PPO, and public RLHF checkpoints that we found for Anthropic use undocumented training splits and/or skip the SFT phase (e.g., the reciprocate/ppo_hh_gpt-j checkpoint you mentioned) which is non-standard and makes fair comparison with DPO difficult.

[1] Scaling Laws for Reward Model Overoptimization. Gao, Schulman, Hilton.

[2] AlpacaFarm: A Simulation Framework for Methods that Learn from Human Feedback, Dubois, et. al.


## Re: Details about IMDb sentiment task

We provide additional details for the IMDb sentiment task, and will revise Section 6.1 to include these details in the next revision of the paper. Following the Transformers Reinforcement Learning (TRL) library, the prompts are prefixes from the IMDB dataset of length 2-8 tokens. We use the pre-trained sentiment classifier "siebert/sentiment-roberta-large-english" as a ground-truth reward model and GPT2-large as a base model. We use these larger models as we found the default ones to generate low-quality text and rewards to be somewhat inaccurate.  We first use supervised fine-tuning on a subset of the IMDB data for 1 epoch. We then use this model to sample 4 completions for 25000 prefixes and create 6 preference pairs for each prefix using the ground-truth reward model. The RLHF reward model is initialized from the GPT2-large model and trained for 3 epochs on the preference datasets, and we take the checkpoint with the highest validation set accuracy. The “TRL” run uses the same hyper-parameters as the TRL library implementation. Our implementation uses larger batch samples of 1024 per PPO step.

---

> ### Comment · Reviewer_FksS · 2023-08-13
> **Followup question**
>
> Thank you for the extra experiments and details.
>
> To clarify, for IMDB, are you reporting reward from the ground truth reward model or your learned reward model? (I assume the former)

---

> > ### Author Response · Authors · 2023-08-13
> > **IMDb reward model**
> >
> > Yes, the evaluation is reported using the ground truth reward model.

---

### Decision · Program_Chairs · 2023-09-21

**Decision:**

Accept (oral)

**Comment:**

Reviewers unanimously agree that the paper addresses an important problem and timely gives an excellent alternative to RLHF to LLMs. Theoretical analysis and thorough experimental results (plus the additional experiments in the rebuttal period) make the paper even stronger. The paper potentially has large impact in the LLM community. Therefore I recommend acceptance as an oral.